# Trusted Operation of Cyber-Physical Processes Based on Assessment of the System’s State and Operating Mode

**DOI:** 10.3390/s23041996

**Published:** 2023-02-10

**Authors:** Elena Basan, Alexandr Basan, Alexey Nekrasov, Colin Fidge, Evgeniya Ishchukova, Anatoly Basyuk, Alexandr Lesnikov

**Affiliations:** 1Institute for Computer Technologies and Information Security, Southern Federal University, Chekhova 2, 347922 Taganrog, Russia; 2Faculty of Science, Queensland University of Technology (QUT), Gardens Point Campus, Brisbane, QLD 4001, Australia

**Keywords:** cyber-physical system, unmanned vehicle, cyber-attacks, anomalies, verification

## Abstract

We consider the trusted operation of cyber-physical processes based on an assessment of the system’s state and operating mode and present a method for detecting anomalies in the behavior of a cyber-physical system (CPS) based on the analysis of the data transmitted by its sensory subsystem. Probability theory and mathematical statistics are used to process and normalize the data in order to determine whether or not the system is in the correct operating mode and control process state. To describe the mode-specific control processes of a CPS, the paradigm of using cyber-physical parameters is taken as a basis, as it is the feature that most clearly reflects the system’s interaction with physical processes. In this study, two metrics were taken as a sign of an anomaly: the probability of falling into the sensor values’ confidence interval and parameter change monitoring. These two metrics, as well as the current mode evaluation, produce a final probability function for our trust in the CPS’s currently executing control process, which is, in turn, determined by the operating mode of the system. Based on the results of this trust assessment, it is possible to draw a conclusion about the processing state in which the system is operating. If the score is higher than 0.6, it means the system is in a trusted state. If the score is equal to 0.6, it means the system is in an uncertain state. If the trust score tends towards zero, then the system can be interpreted as unstable or under stress due to a system failure or deliberate attack. Through a case study using cyber-attack data for an unmanned aerial vehicle (UAV), it was found that the method works well. When we were evaluating the normal flight mode, there were no false positive anomaly estimates. When we were evaluating the UAV’s state during an attack, a deviation and an untrusted state were detected. This method can be used to implement software solutions aimed at detecting system faults and cyber-attacks, and thus make decisions about the presence of malfunctions in the operation of a CPS, thereby minimizing the amount of knowledge and initial data about the system.

## 1. Introduction

Cyber-physical systems (CPSs) are widely used today [1]. CPSs include the Internet of Things (IoT), big data, unmanned vehicles, intelligent systems, etc. [2]. However, various types of influences on the CPS or random component failures can lead to disastrous consequences. Autonomy is a key feature of a CPS, and the results of its actions affect the physical world. CPSs directly impact manufactured products, the environment, and humans [3]. An unmanned vehicle (UV) is a typical example of such a CPS. As a rule, the behavior of an unmanned vehicle is characterized by the various “modes” in which it operates, depending on the tasks assigned to it [4]. In each such mode, a different state-driven “control process” is used to read sensor values and send signals to the actuators.

Typically, a CPS has a multi-level structure. Sensors and actuators are located at the lowest “physical” layer [5]. A sensor differs from an actuator in that the sensor passively receives and sends information while performing measurements [6]. Thus, a sensor does not affect the environment and only receives information about it. At the same time, an actuator activates some mechanisms by receiving a control signal, i.e., it actively interacts with the environment. In the case of a UV, examples of sensors are a compass, a barometer, an altitude sensor, etc., and electric motors can serve as actuators [7].

As a rule, apart from the physical layer, most researchers distinguish between three other layers of a CPS [8,9,10]. The physical layer often includes not only sensors and actuators, but also microcontrollers. In the model developed herein, we attribute microcontrollers or single-board computers to a separate layer. These devices are layered between the sensors and the control layer [11]. From the point of view of the functional model, they perform the function of aggregating and transferring control actions.

An important layer between the sensors, microcontrollers, and the control layer that is distinguished by all researchers is the communication network components. For our functional model, this layer is not singled out separately, but is combined with the control level [12]. Thus, we separate high-level control, which includes a human–machine interface, data storage, artificial intelligence, automation algorithms, specialized control software, and in particular, a system state analysis machine. Low-level control includes microcontroller firmware, the microcontrollers themselves, and the algorithms and software used for controlling the sensors [13].

A generalized scheme for representing the CPS layers in terms of functional features is shown in Figure 1. In addition to the sensors, detectors are separately presented at the physical level to highlight the type of sensors triggered by mechanical pressure. Various buttons and opening and closing valves are examples.

Our research concerns verifying that a CPS is in the correct operating mode as a way of detecting anomalies. Choi et al. proposed using a digital twin model to solve the problem of CPS verification [14]. They perform digital 3D modeling, as well as modeling the processes and data in the system. Their integrated platform enables data analysis and fault detection.

Yang et al. proposed a system for analyzing errors in the operation of a mobile robot for detecting anomalies [15]. They offer a way to detect hidden anomalies in a robotic system by collecting sensor and actuator data for analysis. They present a residual-based method that can detect attacks (zero-dynamic attacks, stealth attacks, and replay attacks) directed at mobile robots that cannot be detected using a normal residual signal. First, the hidden conditions of the three attacks are explained and defined, and their theoretical representations are given.

Building on this previous research, we aimed to improve the efficiency of the methods used for detecting faults in the operation of a control system and develop methods for detecting anomalies based on analysis of the system states. Such a capability allows decision making to improve the CPS’s fault tolerance. To do so, it is essential to identify the set of processing states in which the CPS can operate and the conditions under which it can transition from one state to another one. The overall behavior of a control system is often called its operating “mode”, and the finite state automaton defining when the transitions between these modes are possible is called a “mode machine” [16]. In each such mode, the system executes a different control process, which itself consists of a cycle of read–process–write states. The behavior of the control process may be defined either as a set of input–output equations or a finite state machine.

Trust in an automated system is an important factor in its development and commissioning [17]. In addition, it is important to consider the factors by which credibility of the system is assessed. The level of performance exhibited by a system can affect the credibility of that system. Thus, the proper design of an autonomous system can be aided by measuring the trust in such systems. Lochner et al. added psychophysical factors to the assessment of trust in systems [17]. They took into account the state of the operator who controls the UAV as a trust factor by analyzing their galvanic skin response (GSR). In addition to the standard metrics, which are obtained by interviewing the operator, they collected skin conductivity data using a commercially available GSR (Shimmer Sensing) sensor. This device was attached to the wrist of the left hand, from which sensory electrodes were attached to the palmar surface of the index and middle fingers using two wires. The device transmitted data via Bluetooth to a laptop for data collection. This approach is quite interesting, but not entirely applicable here because modern UAVs for the most part should be autonomous. In addition, modern UAVs have built-in functions in the flight controller, which at the stage of its operation, do not allow a person to fully influence the flight process, but only allow you to give certain sets of commands.

Keshavarz et al. [18] proposed a trust monitoring mechanism, in which the ground station continuously monitors the behavior of the UAVs in terms of their trajectory, the energy they consume, as well as the number of tasks they have completed, and evaluates the UAV’s trust level to detect any abnormal behaviors in real time. Their simulation results show that the trust model can detect malicious UAVs that can be subject to various cybersecurity attacks, such as flooding attacks, man-in-the-middle attacks, and real-time GPS spoofing attacks. When they were modeling a GPS spoofing attack, the authors presented a drone flight graph. However, their system may be ineffective since the base station can also be attacked and is a single point of failure. In addition, the CPS node may be out of the field of view of the base station. Our approach, in contrast to this study, allows the CPS node to independently make decisions about the trust in the control processes.

Barka et al. proposed a new trust-monitor-based communication architecture for Flying Named Data Networking (FNDN) [19]. First, monitor nodes are selected based on their trust and stability. The monitors then become responsible for propagating the data packets to avoid the broadcast storm problem. At the same time, intermediate UAVs choose whether to validate the data or not, following their subjective opinion about the behavior of their manufacturer, and thus reducing the computational complexity and latency. The simulation results show that this work can support security levels in excess of an 80% dishonesty detection rate. The main idea is to establish trust between the interacting UAVs and to select the most reliable UAV (called a monitor) with a high probability of storing data and having enough power for the rest of its mission. Their study is based on the control of the UAV actions, but they describe the categories of actions poorly. In addition, the presence of monitor nodes carries additional threats, since there are attacks that are aimed at impacting the trust. Their basic question is about the lifespan of the UAV network and how to assess trust if the UAV has just arrived in the group. On the one hand, it will have a lot of energy, but on the other hand, it will not have any historical recommendations. In contrast to their study, our approach allows the node to act autonomously and correlate its expected behavior with the behavior of the group on its own, which increases the degree of autonomy, and hence, the reliability.

A new context-sensitive trust-based solution for distinguishing between intentional and unintentional UAV anomalous behaviors has been proposed by Barka et al. [20]. The method simultaneously establishes trust between the UAVs and evaluates the current context in terms of UAV energy, mobility pattern, and queued packets. When they were evaluating the confidence, the UAV evaluated the buffer occupancy, energy, and mobility patterns of the UAV. Thereafter, if the system detects that any adjacent UAVs have inadvertently dropped packets, it adds a confidence correction factor to the overall inter-UAV trust calculation. To assess the mobility of the UAV and decide whether the UAV is unintentionally dropping packets, a channel stability index is calculated. An advantage is that it normalizes the parameters and brings them to values from 0 to 1. However, the main calculations are related to the transmitted and received packets and the communication channel. Such methods will not be suitable for cases where the UAV operates autonomously, and it will also be difficult to detect a data spoofing attack. In contrast to their method, in our approach, the node relies on the readings of internal sensors, and its estimates can be built on the basis of a combination of readings from any sensors at the physical or firmware levels.

Maalolan [21] considered the assessment of trust in UAVs in a way that is similar to our method herein. He also considered the behavior of UAVs in different modes. There was an assessment of the flight speed, the rate of climb and descent, the quality of flying around obstacles, and more factors. Maalolan also considers the need to estimate the growth rate of the parameter in certain regimes, as we also propose in our study. A feature of his work is evaluating the raw data. In our study, we use normalization and evaluate confidence in terms of the probability. In addition, Maalolan used monitors for each mode and set of metrics, but the final value of trust in the system was not developed.

Singh and Verma [22] believe that since UAVs cooperate and coordinate with each other when they are performing a mission, trust between the nodes (UAVs) is a critical aspect. They proposed a trust model in which a genetic algorithm is used to optimize the weights of various parameters to estimate direct trust values. Direct trust is combined with recommendations to calculate the final host trust value. To evaluate the trust of a node, three reliability attributes and one performance attribute are used. Node energy, signal strength, and the packet delivery ratio reflect the node reliability, while transmission delay reflects the performance. Based on the observed parameters, direct trust is calculated by aggregating the various parameters using their optimal weights.

Mohammed et al. [23] considered a group of UAVs as a Mobile Ad Hoc Network (MANET), where individual UAVs are modeled as nodes. They discussed various trust-based protocols and control schemes that can be used in UAV networks and define the areas of UAVs in which such protocols can be applied. Trust management has a variety of uses in many decision-making situations, including intrusion detection, authenticated access control, key management, the isolation of misbehaving hosts for efficient routing, and other purposes. However, the use of cryptographic techniques to validate trust may not be appropriate for a CPS. This is because an attacker can take over control of the node. That is why the node itself must analyze the changes in processes and evaluate its trust in them. If a node is captured by an intruder, it must determine this itself and implement countermeasures, as is possible in our approach.

Birnbaum et al. [24] created a prototype of a UAV monitoring system that collects flight data, and then evaluates and monitors airframe and controller parameters in real time. They focused on three types of attacks: hardware failure, malicious hardware, and attacks on flight control computers. By monitoring the parameters of the flight controller, it is possible to determine whether the controller is behaving in accordance with the design specifications or not. For the flight model, the authors chose to monitor all of the available flight data, including information about the roll, pitch, yaw, elevator, throttle, rudder, aileron, and GPS position. Applying the recursive least squares (RLS) procedure to the flight data yielded a sequence of parameter estimates that fluctuated over time due to measurement noise and a number of unaccounted for factors. These estimates are randomly distributed around some unknown True values that represent the UAV’s current state to identify and check the deviations between the direct (calculated) and nominal values of the parameters known to the UAV’s developers. By contrast, however, the advantage of our approach is that knowledge of the normal values can simplify calculations, but it is not required. In addition, our method evaluates the dynamics of the process and controls the behavior of the process in different modes of operation, which increases the accuracy of the assessment when we are determining anomalous behavior.

Stracquodaine et al. [25] proposed a comprehensive system for protecting UAVs from hardware and software attacks at the lowest and highest semantic levels. This approach monitors the autopilot directly, as well as the lowest level of the operating system on board the Unmanned Aerial System (UAS), to determine whether the malware has altered the aircraft’s functions or not. Related events are used to produce directed graphs, which in turn are compared with each other. A set of unique graphs obtained during the normal operation of the system constitutes a profile. This profile can be used to track the abnormal behavior while the UAS operates. However, a disadvantage of their approach is that the event graph can be quite large, and this indicates that before the system was launched, a detailed analysis of it and full control of all of the possible transitions must be carried out. Such a system cannot be applied to different types of UAVs and cyber-physical systems. In contrast to their approach, we rely on process control at a higher level and describe the process by changes in the cyber-physical parameters, which makes it possible to carry out analysis by mathematical methods.

Ogden et al. [26] presented a way of applying multidimensional statistical process control methods to improve and automate anomaly detection in mechanical systems. Examples of their implementation for detecting operational anomalies in mechanical systems are, for example, the operation of a turbocharger, wind turbine generator failures, the analysis of transients in the operation of an aircraft jet engine, or monitoring the performance of an air compressor. The methodology of multidimensional statistical process control reveals anomalies in the complex relationships in the sensor data collected during the operation of a mechanical system. This work is aimed at assessing anomalies in the operation of mechanical systems. This work has similarities with ours, since the mechanical process that the authors consider is similar to the cyber-physical processes. However, the authors write that their process requires a large amount of data and requires precise thresholding to reduce the number of false positives. For our approach, the thresholds are determined based on methods of probability theory and do not require a large adjustment.

Stojanovic et al. [27] proposed the use of a multivariate approach for anomaly detection based on comparison with the collected normal data. Their approach to multivariate anomaly detection is based on modeling and managing the flow of variations in a multidimensional space. A feature of the approach is the ability to observe relationships between the variations in a large set of parameters and to create clusters of “normal/usual” variations. To achieve scaling, which is one of the most complex requirements, the approach is based on the use of big data technologies to implement data analytics tasks/calculations. Their approach typically requires higher computational complexity than univariate or model anomaly detection does because the underlying algebra used to calculate the multidimensional anomalies is computationally intensive.

Sabo and Cohen [28] present a decision-making method for flying around obstacles using fuzzy logic rules. Their method allowed them to convert binary parameters into fuzzy logic parameters. The method is quite effective for such a task. Nevertheless, in our study, it is important for us to detect not only the presence of some changes, but also to understand the nature of the process in order to verify the significance of the changes with respect to trust in the process. The metrics we have proposed allow us to do this.

Sun et al. propose a method for analyzing the failure of the GPS system in UAVs [29]. Their method, according to their authors, can be trained during operation. However, the method requires a database for training. In this case, the data must be marked up for storage and a considerable volume of data is required to implement the method. By contrast, in our solution, prior data are needed only to confirm the effectiveness of the method and to verify it.

Not all cyber-physical systems are static and require a large amount of computing resources. In the case of UAVs, even under a navigation signal spoofing attack, the latitude, longitude, and altitude may not necessarily change simultaneously (as is also shown in our study below). They depend on the data that the attacker forged. In addition, the failure of one sensor, such as an altitude sensor, will not affect speed and positioning, but it will affect the accuracy of determining the altitude. In the case of a UAV, tracking each parameter separately and analyzing the process of its change are also advisable because the UAV is itself a highly sensitive system and is affected by environmental conditions. At the same time, various sensors can change the readings for the same kind of attacks, and vice versa, the same parameters can change for different attacks. Therefore, it is important to investigate the nature and extent of these changes for further classification. In addition, the multivariate approach requires significant computing power [27], which the UAV often does not have.

In this connection, the approach we develop here moves away from the need to collect data sets from normal flights and use them to estimate the confidence state. Our approach instead aims to minimize the amount of prior knowledge and initial data needed to analyze the system effectively. This generalizes the approach and supports the fact that there are many UAV manufacturers using different technologies.

## 2. Materials and Methods

### 2.1. Functional Model of a Cyber-Physical System

If the CPS of interest is a UV, data sets that can be received from the physical layer and go to higher layers travel via the flight controller. The flight controller belongs to one of many microcontrollers $MC=\left\{mc_{0},\ldots ,mc_{i}\right\}$, which are a subset of *LLC* low-level control, so the notation is:(1)$$mc_{fc,i}\in \;\left\{LLC\right\}\;,$$
where $\left\{LLC\right\}$ is the set of components that are a subset of low-level control.

Since the flight controller, in fact, includes both the microcontroller itself and the set of sensors that are connected to it and from which it receives information that it transmits further and it can also transmit information to the actuators, we can say that sets of low-level control and physical layer objects can intersect, and such an element as a flight controller belongs to both sets:(2)$$LLC\cap PH=\left\{mc_{fc,i}|mc_{fc,i}\in \;LLC,mc_{fc,i}\in PH\right\},$$
where *PH* is the collection of physical layer components.

Accordingly, we assume that the flight controller is also characterized by a set of cyber-physical parameters that it gives to a higher layer. Moreover, it can receive cyber-physical parameters from several elements of the set {PH} at once. Thus, a vector of parameters is formed, which may include a plurality of cyber-physical parameters of the flight controller.

Some cyber-physical parameters can be obtained from different sensors, for example, the flight altitude value can be obtained from GPS or LIDAR. At the same time, such values obtained from different sensors may not coincide due to errors or inaccuracies in the operation of the systems [30]. Therefore, in this study, even though various devices may produce the same knowledge, in this study, we will consider them as separate cyber-physical parameters. Thus, the cyber-physical parameters enter the high-level control component, where they are analyzed and determined in accordance with the expected parameters.

Let us consider a functional model of a CPS using the example of an autonomous or unmanned vehicle. Such an example combines several concepts of CPSs at once, such as autonomy, intelligent control, sensor networks, distributed computing, and wireless technologies.

The following concepts are the main issues when one is building a model. A mode of operation is a set of conditions and operations (described by the documentation or determined based on the tasks performed by the system) that the system performs in accordance with an established algorithm. A process is a repeatable sequence of actions and modes, which lead the system to achieving an established goal.

Let us describe the operating modes of the system. Mainly, any cycle of operation of a CPS can be framed by switching it on and off. As a rule, the process of turning on a CPS is due to a poll of the components in the physical layer. When one is polling components, as a rule, low-level control components receive values of True or False.

Depending on the mode, the normal behavior of a UV is considered to be a change in some of the telemetry parameters defined for each of the modes. Thus, the turn-on mode $om_{0}$ is characterized by polling the components $CP_{i,ph}$ in the physical layer and achieving values of True for each one. After that operation, the CPS could transition to the next mode (the terms in these equations are described in more detail in Table 1 below):(3)$$\left\{\begin{array}{l} om_{0}=f_{bul}\left(CP_{i,ph}\right)\\ om_{0}\to Tr \end{array}\right.$$

After this special initialization mode, the transition comes to a standard operating mode. After polling the sensors, the system should receive a task for execution, and only then will the main workflow begin. A normal operating mode is determined by the CPS fulfilling any tasks received from the operator.

Typically, the transition from the turn-on mode to the normal operation mode may be accompanied by a transition to the standby mode $om_{1}$. For an unmanned aerial vehicle (UAV), for instance, this mode can be referred to as HOLD ON GROUND. In this mode, the CPS does nothing, it only passively waits for a command from the operator. At the same time, data from the sensory system are already coming in continuously. That is, the standby mode is accompanied by the process of data exchange between the CPS subsystems, as well as between the operator and the CPS. Thus, the mode considers not only the state of the cyber-physical parameters that come from the sensors $CP_{i,ph}$, but also the cyber-physical parameters of the network layer $CP_{i,nc}$:(4)$$\left\{\begin{array}{l} om_{1}=f_{tr}\left(CP_{i,ph},CP_{i,nc}\right)\\ om_{1}\;\to Tr \end{array}\right.$$

In this mode, a slight change in the GPS coordinates (latitude, longitude, and altitude) is considered as normal because a UAV moves a little when it is trying to maintain its current position; the angle of the motion direction (heading_deg), the angle of yaw, and the corresponding minor changes in speed and acceleration in the directions of *x*, *y*, and *z* can also change slightly. Deviations within 0.5–1.5 m are implied. In this case, measurements of change have been identified experimentally based on many trials, as well as the results of theoretical analyses.

In this case, a deviation from the normal state without the transmission of control commands will be an explicit movement of the UV, which can be determined by a strong change in the coordinates.

The flight mode MANUAL is denoted $om_{1,man}$. As a rule, the UAV is in this mode at the start of the mission before the beginning of autonomous control or artificial intelligence operation, namely, before the transmission of the first control command. In this case, the UAV should not move, the engines are turned off, and the vehicle itself is on the ground. If it is in this mode in the air or during a mission, this is a clear deviation from the normal operation, as this indicates a transition to manual control.

After a command transmission from the operator or by another trigger is received, the CPS switches to the operating mode $om_{1}$. In this mode, the system begins to perform a set of actions $A=\left\{a_{0},\ldots ,a_{i}\right\}$ provided by the specifics of its operation. Typically, a CPS will perform a different set of actions in a different order during each of its operating modes. The sequence of actions is determined by the control process that is in effect during that mode. The order of actions, which is determined by the algorithm or controlled by the operator, artificial intelligence, etc., should lead the system to the fulfillment of a target goal $T=\{t_{0},\ldots ,t_{i}\}$. The number of elements in the target set depends on the capabilities and purpose of the CPS. The normal operation mode for the CPS can be equated to the mission mode. If we are referring to an aerial vehicle, then we introduce an additional takeoff mode for the UV.

In the mode TAKEOFF $om_{1,t}$, it is normal to change the latitude and longitude parameters slightly and for the altitude parameter to change significantly, since in this mode, an aerial vehicle takes off from the surface and gains its assigned altitude or begins to move. In this case, the parameters of speed and acceleration in the directions of *x*, *y,* and *z* can also be changed accordingly. After transmitting the command TAKEOFF, the CPS will automatically enter the flight mode HOLD. The instantaneous occurrence of flight mode HOLD is possible even if other control commands have been transmitted. The CPS in this stage may not be exchanging data with the operator since it is already starting its mission. In this stage, it is important to track any change in the parameters and detect the change in the altitude. So, the conditions for this mode are as follows:(5)$$\left\{\begin{array}{l} om_{1,t}=f_{tr}\left(CP_{n,ph}\right),\;f_{pois}\left(CP_{n,ph}\right),\\ om_{1,t}\;\to 1. \end{array}\right.$$

At the same time, a deviation from the normal state without the transmission of control commands will be a clear significant deviation of the UAV from the coordinates or take off to a height that is greater or less than that which was transmitted in the control command, taking into account the errors (0.5–1 m in latitude and longitude and 0.5–1.5 m in altitude).

A flight mode HOLD with movement $om_{2,h}$ may occur if the control command GO_TO has been issued. Normal behavior in this mode will be considered as moving the UAV along the coordinates and altitude specified in the command. Flights at a different altitude or movement along other coordinates or a lack of response to a transmitted control command will be considered as deviations.

In the mode MISSION $om_{2,m}$, the UAV performs the task given by the operator. The behavior in this mode is considered to be normal when there is a change in telemetry in terms of coordinates, yaw angles, and acceleration in accordance with the transmitted control command, which contains all the mission points and parameters for these points. There may be a slight change in the altitude and speed depending on the task.

Thus, to establish a stable state of flight in this mode, it is necessary to check the parameters for compliance with the flight task. In addition, the tracking of changes in $CP_{i,ph}$ is needed. If $CP_{i,ph}$ does not change, it means that the UAV does not move, which may be not typical for this mode. In this case, it depends on the flight task significantly.

Moving to other coordinates or waiting too long at one of the waypoints (the time spent at the waypoint is also set in the flight task) will be considered as abnormal. Accordingly, states when parameter $CP_{i,ph}$ does not change for a long period will be considered as suspicious and unstable.

After completing the mission, the UAV should land or slow down and stop. At the same time, there are two options for the return of the UAV: the normal mode, when the UAV has completed the mission and landed, and the emergency mode, when the UAV for some reason could not complete the mission and returns to the takeoff point.

In the mode Return to launch (RTL) $om_{3,rtl}$, the UAV returns to the takeoff point. This mode is characterized by the fact that the UV chooses the shortest route from its current waypoint to its home location. The UAV decreases its altitude after returning to the starting point. Accordingly, there is a change in the coordinates and a decrease in the altitude.

In the mode LAND $om_{3,l}$, the UAV decreases its height. This mode is characterized by the fact that the flight altitude decreases, while the coordinates remain unchanged.

The special Shutdown mode is performed manually by the operator when the system is completely de-energized and all of the sensors have been turned off.

### 2.2. State Machine of a Cyber-Physical System

In general, a UV performing tasks in autonomous mode is accompanied by a subdivision of many processes into sets of subprocesses. Each of the processes is associated with the CPS’s operating modes if they are explicitly set in the system. A comparison of the UAV processes and modes is presented in Table 1.

When one is describing the operation of a UV, it is important to consider specific control processes and modes, as well as their sequences. Figure 2 shows a diagram of the change in the processes and their relationship with the modes for our UAV example. Control processes are represented by ovals, and modes are denoted by rectangles. The two green rectangles indicate operator actions that are performed manually.

As long as it is in a particular mode, the system will perform a corresponding control process. This may be a periodic process, in which the UV samples the sensors and adjusts the actuator settings at regular intervals, or it may be a sporadic process which responds to various triggering events, either internal or external ones. Either way, the process continues the execution as long as the system remains in the current mode. However, when something causes the system to change the mode, a different control process will be initiated for the duration of the new mode. Thus, each operating mode normally corresponds to one control process, and whichever mode the system is in determines which control process is currently being executing.

The CPS must be physically launched by the operator. The purple ovals indicate processes that are weakly related to modes. Here, the sensory system is checked, and the possibility of functioning is assessed or, vice versa, it is turned off and brought into a state of inoperability. This scheme must be taken into account in order to determine the possibility of the system transitioning to a new state.

These processes are basic and can be mapped to many states. That is, each of the processes can be in one of the following states when they are being executed:-A stable state $S_{0,st}$, which is when the UV performing actions $A=\left\{a_{0},\ldots ,a_{i}\right\}$ within the framework of a legitimate process approaches with a high probability the fulfillment of the goal $T=\{t_{0},\ldots ,t_{i}\}$ set by the task;-An uncertain state $S_{1,uc}$, which is when the UV performing actions $A=\left\{a_{0},\ldots ,a_{i}\right\}$ approaches the boundaries of the current process, and the probability of achieving the goal $T=\{t_{0},\ldots ,t_{i}\}$ set by the task decreases;-An unstable state $S_{2,us}$, which is when the UV performing actions $A=\left\{a_{0},\ldots ,a_{i}\right\}$, can go beyond the scope of the current process, and there is a low probability that it can achieve the goal $T=\{t_{0},\ldots ,t_{i}\}$ set by the task;-An under-stress state $S_{3,a}$, which is when the UAV performs actions $A=\left\{a_{0},\ldots ,a_{i}\right\}$ incorrectly, going beyond the scope of the current process, with a high probability that it cannot achieve the goal $T=\{t_{0},\ldots ,t_{i}\}$ set by the task.

A process ${\mathrm{PR}}_{i}\;\in \left\{PR\right\}$ is trusted when it is in the correct state. It is possible to move from process ${\mathrm{PR}}_{i}$ to process ${\mathrm{PR}}_{j}$ if the former process was in state $S_{0,st-1}$ and the new process is in the stable $S_{0,st}$ or uncertain state $S_{1,uc}$ and it does not contradict the scheme of possible mode transitions and the scheme of process transitions.

An unstable state may occur as a result of deviations in one or two parameters. Then, the confidence value will be slightly lower than 0.5. This situation is possible in the case of some deviations in flight, for example, due to the wind, when the drone cannot maintain the desired speed or altitude. The state under stress occurs due to significant external factors, usually because of an attack. Then, three or more parameters are changed, but as the study shows, all of the parameters change.

Thus, when one is evaluating the state of the currently executing process, it is important that it changes within the allowable values and brings the system closer to achieving the successful execution of the current task. The scheme of modes and the determination of the possibility of a mode change are two of the metrics of state verification.

Figure 3a shows the relationship between the cyber-physical parameters, metrics, and operation modes. Metrics are understood as functions that normalize the parameters and allow them to be used as a measure of assessing the degree of trust in the system’s overall integrity.

The metric and mode of operation are inputted into the state analyzer, which determines the degree of confidence in the UV’s state in real time and verifies the process. If the current process is trusted, then the system can move to another process, as shown in Figure 3b by the green arrows. The red arrows show possible the transitions when the system is affected by an anomaly, which could be caused by a component fault or a deliberate attack. Figure 3 shows that if the system is not behaving correctly, it will immediately stop without completing the current task. This transition is the most critical one.

At the same time, it is not necessary that an anomaly will lead to the emergency termination of the process; the system may try to continue executing the process, but with limited capabilities.

States can be expressed as a probability function based on the above definitions. To determine the possibility of transition of a CPS from one state to another state, a set of metrics presented in Table 2 was used.

Metric 1, the *reliability of performed functions in the current state*, allows us to determine which current parameters of the CPS correspond to the given boundaries. It is necessary to evaluate whether the process being implemented goes beyond the allowable interval or not.

When one is determining the boundary of the confidence interval in the presence of target indicators, the target indicators are the values of cyber-physical parameters that the CPS must achieve. For instance, in a UAV example, the altitude and speed are determined by the flight task. In addition, for most UAVs, the lower and upper limits of the flight altitude are set in the firmware [31].

The boundaries based on knowledge of the CPS are calculated based on knowledge about the reference behavior of the system, which can be obtained as a result of theoretical calculations, original system specifications, and/or modeling.

The boundary of the confidence interval based on the previous values from the sample is determined in the case when there is no input information about the normal performance of the system and information about the reference behavior of the system. In this case, the boundaries of the interval can be calculated dynamically. To do this, it is necessary to build a confidence interval based on the previously observed behavior of the system.

Metric 2, the *control of increase/decrease in cyber-physical parameters*, allow us to assess the presence of changes during the implementation of the CPS process. In some cases, an improvement in the performance of a parameter should be observed. In the UAV example, this could be during takeoff, or vice versa, a decline is observed when it is landing. During takeoff and landing, it is necessary to monitor the changes in cyber-physical parameters, in particular the altitude and flight speed, otherwise the expected phenomenon will not be observed [30].

Metric 3, the *trust in the current process.* When one is assessing the state of the process, it is necessary to consider what mode the system is in, which modes it can go into, and which ones it cannot go into. If the system does not explicitly track the modes, then control commands can be used to determine what the system should do.

For example, in our UAV case study, the takeoff mode condition looked like this:(6)$$\left\{\begin{array}{l} f(om_{i})=1,\\ f_{tr}(CP_{i,\min}<CP_{i}<CP_{i,\max})>0.5,\\ f_{bool}(CP_{ia})=1,\\ f_{pois}(CP_{i}|\;\bar{CP_{i}})\;>0.5,\\ f_{pois,tr}(CP_{i,\max}|\;CP_{i,\min})\to 1,\\ Tr>0.6;Tr\neq 0. \end{array}\right.$$

The conditions for the return to the starting point and landing mode were the same, except for the value of the function $f_{bool}(CP_{ia})$, which changed its value to 0.

Thresholds are taken based on the definitions of probability theory and distribution types. The threshold for the cumulative function is chosen because if the parameter does not deviate from the mean or expected values, then the distribution is 0.5. If a deviation occurs, then the value tends to be 0 or 1. Since the experimental study shows that during normal flight some deviations from the course are possible due to wind or GPS inaccuracies, we have set an acceptable deviation of 10–15% and thresholds of 0.4–0.6 for the cumulative function. To obtain the confidence interval, Metric 1 should tend to 1. If the threshold of 0.5 is crossed, then we can consider trust. Usually state 0 is considered to be untrusted, and 0.5 is undefined, as also described by Keshavarz et al. [18].

### 2.3. Development of a Software Module for A State Analysis Machine

Our implementation of the experimental software module uses the Python programming language Version 3.8. To transfer data from the sensors and transmitters, a message broker of the MQTT (Message Queuing Telemetry Transport) protocol was used. The developed architecture uses the event-driven approach (event-driven architecture).

The event-driven architecture pattern is a popular distributed asynchronous architecture pattern used to build applications. This is a modern approach to design based on the data describing “events”. Event-driven architecture allows an application to respond to these events as they occur. The main benefit of an event-driven architecture is responsiveness. Since everything happens in real time, this architecture provides the fastest response time.

The event-based approach has become extremely popular in recent years due to both the great growth of data sources that generate events (IoT sensors) and the development and adoption of technologies that process the flow of events, such as Hazelcast Jet and Apache Kafka.

An event in such a system is a message that represents the data or commands when an observed value changes, such as an increase or decrease in the flight vehicle altitude. This message or event is generated by what is called an event emitter, which is an entity that detects a change and notifies the system.

In our prototype, the sensors of the flight vehicle are the event generator. A common design pattern that is used to implement this process is the publish/subscribe pattern. In this case, the event emitter is called the publisher, and the stakeholders are called event subscribers or event handlers.

MQTT is one of the most widely adopted IoT communication protocols that supports the above architecture and is based on a publish/subscribe communication pattern using an MQTT broker to coordinate event delivery. The broker will receive events from the producer and forward them to the subscribers.

As part of the development, a specification for data transfer via the MQTT protocol was designed. Table 3 lists the main topics for obtaining data from a flight controller.

The general structure of the data acquisition methodology is shown in Figure 4. The responsibilities of the elements are as follows:-UAV Worker—the main class for launching the creation and initialization of all of the objects of other modules;-FSM (Final State Machine)—responsible for the context in which the UAV operates, stores the state of the cyber-physical system, and meets the conditions for transitions between states;-State—the base class for describing the state;-Transition—the base class for transitioning and running checks of the conditions of the transitions between the states;-Model—the base class of the context in which the UAV operates;-Callback—the base class for how the callback methods work;-Validator—the base class of the transition condition;-MQTT Client—the module responsible for receiving data from sensors and transmitting control commands to the flight controller.

### 2.4. Algorithm of the State Machine for Determining the State of the UAV

Our state machine for a UAV aims to track whether the process under observation goes outside the confidence interval. In addition, it must assess whether the process is changing according to the expectations. For this, the metrics presented above were used. An estimation of the probability of the values falling within the boundaries of the confidence interval is made if the parameter exceeds a threshold of 0.5 because the parameter’s value varies from 0 to 1; the closer to 1 it is, then the greater the probability is of it falling into the confidence interval. Given the allowable deviations, a value above 0.5 is acceptable; this threshold is set when one is creating a software module that is unchanged.

Next, it is important to evaluate the increase of decrease in the parameter values. For the takeoff process, it is important that the acceleration and altitude (and in some cases, also the latitude and longitude) increase. The increase in the value is determined by the Cumulative Poisson function. It is designed in such a way that when there is an increase in the value, then there is a deviation from the average value upwards, and thus the probability that the function reaches the “desired” value (that is, one which is above the average value and close to one) increases. Therefore, the improvement of the parameters can be determined by increasing the Cumulative function above a threshold of 0.6. If there is a decrease in the value, and it becomes less than the average value (or expected value), then the probability of it reaching the “desired” value decreases, the value of the function tends to zero, and it will become less than 0.4. This situation is normal when the aircraft is landing or the system is turned off. These thresholds are also embedded in the software module. The software module works according to the following algorithm.
Determine the current flight mode.
1.1If the UAV is in the current flight mode and switches to a new flight mode, then determine whether this transition is legitimate according to the transition graph, as shown in Figure 2 and Figure 3.1.2If the mode is legitimate and the transition to the mode is legitimate, then $f(om_{i})=1$, if not, then it becomes 0.Determine the reliability of the functions performed in the current state.
2.1From the flight task file, it is necessary to take information about: the altitude, flight speed, longitude and latitude, their maximum, and minimum values and waypoints.2.2Calculate the boundaries of the confidence interval using formulas from lines 1.1.1 and 1.1.2 in Table 2.2.3From line 1.2.1 in Table 2, calculate the function of the probability of falling into the confidence interval for each of the parameters.2.4If condition $f_{tr}(CP_{i,\min}<CP_{i}<CP_{i,\max})>0.5$ is met, then set the parameter to 1 and go to Step 3.2.5If condition $f_{tr}(CP_{i,\min}<CP_{i}<CP_{i,\max})>0.5$ is not met, determine the parameter for which this condition was not met and assign the value 0 to the parameter. Return the value of the parameter and go to Step 3.Control the increase/decrease in the cyber-physical parameters (Metric 2).
3.1Calculate the average value of the cyber-physical parameter for $\Delta w_{ij}$ intervals using the formula from line 2.2 in Table 2.3.2Calculate the value of the cumulative Poisson function for each cyber-physical parameter using the formula from line 2.1 in Table 2.
3.2.1If the current processes are $PR_{{}_{2,wait}}^{}$ and $PR_{{}_{3,proc}}^{}$, then $f_{pois}(CP_{i}|\;\bar{CP_{i}})\;>0.6$, or $f_{pois}(CP_{i}|\;\bar{CP_{i}})\;<0.4$ and goes to zero, then set the parameter to 1 and go to Step 4.If the condition above is not met within $\Delta w_{n}$ intervals, then set the parameter to 0 and return the parameter value, for which the condition is not met ($\Delta w_{n}$ set is optional, while the default is 3), and go to Step 4.If the current modes are $om_{i}$ = $om_{1,man}$, $om_{1}$, $om_{2,m}$, $om_{2,h}$, and $0.4<f_{pois}(CP_{i}|\;\bar{CP_{i}})\;<0.6$, then set the parameter to 1 and go to Step 4.If the condition above is not met within $\Delta w_{n}$ intervals, then set the parameter to 0 and return the parameter value, for which the condition is not met ($\Delta w_{n}$ set is optional, while the default is 3), and go to Step 4.3.2.2If the current processes are $PR_{{}_{0,on}}^{}$ and $PR_{{}_{1,start}}^{}$, then the processes change the confidence function (formula from line 2.3 in Table 2) for latitude and longitude calculated relative to the starting point. That is, it is necessary to take the coordinates of the point from which the UAV starts taking off, obtain the lower and upper bounds of the confidence interval, and then calculate the value of the function.If the value for the parameters of altitude and velocity is $f_{pois,tr}$ > 0.6 and tends to 1, then assign a value of 1 and go to Step 4.If $f_{pois,tr}$ < 0.6 during the intervals $\Delta w_{n}$, then assign a value of 0 and return the value of the parameter for which the condition was not met ($\Delta w_{n}$ set is optional, while the default is 3), and go to Step 4.3.2.3If the current processes are $PR_{{}_{4,stop}}^{}$ and $PR_{{}_{5,off}}^{}$, then the process change confidence function (formula from line 2.3 in Table 2) for latitude and longitude calculated based on the point at which the UAV is aiming (also taken from the flight task).If the value for the parameters of altitude and velocity is $f_{pois,tr}$ < 0.4 and tends to zero, then assign a value of 1 and go to Step 4.If $f_{pois,tr}$ > 0.4 during the intervals $\Delta w_{n}$, then assign a value of 0 and return the value of the parameter for which the condition was not met ($\Delta w_{n}$ set is optionally, while the default is 3), and go to Step 4.Calculation of the final value (Metric 3) of trust by Formula 3.1 in Table 2. (For the final trust value, the values of each metric for each parameter are taken, depending on whether the values calculated above tended towards 0 or 1.)
If $Tr(f_{tr}(\{CP_{i}\}),f(om_{i})$ = 0, it is necessary to provide an emergency response.If $Tr(f_{tr}(\{CP_{i}\}),f(om_{i})$ > *threshold* value, it means that the state is stable. (The threshold value depends on how many metrics we allow to be zero. We will assume that if three or more metrics are equal to zero, then an anomaly occurs, so the threshold will be taken as 0.6. This value is optional and can be changed by the user when they are testing the algorithm.)If $Tr(f_{tr}(\{CP_{i}\}),f(om_{i})$ = 0.6, the system is in an undefined state.If $Tr(f_{tr}(\{CP_{i}\}),f(om_{i})$
*<* 0.6, the systems is in an unstable state.

We used a floating window that was equal to three time intervals. If the size of the floating window is increased, then the speed of anomaly detection will decrease, however, if the size is decreased, then the accuracy will be reduced. This value can be changed depending on how often the values are received from the flight controller. In our case, values are obtained three times per second. Based on this, a floating window was also defined. As the frequency of data collection increases, the window may change. These calculations do not affect system performance by themselves due to the presented architecture of the software application.

## 3. Results

To test our method for anomaly detection, we used open data on the flight of a UAV from the public domain [32,33].

Let us consider the flight mode HOLD IN AIR. In this mode, the UAV is in flight while it is performing a mission. In such a situation, a cyber-attack or onboard component failure can have a destructive effect on the UAV because it is outside its controlled zone. As an example of such a destructive impact, we consider a GPS signal spoofing attack. This attack is characterized by the fact that it replaces the UAV’s coordinates. At the same time, the UAV can lower its altitude or change its direction of movement, etc.

When one is calculating the probability that the current value of a cyber-physical parameter falls within the confidence interval for the case when the UAV is moving normally, the indicator fluctuated within 0.6–0.9 for all of the parameters, which is within the confidence interval. The purpose of calculating the confidence interval is to build such an interval based on the sample data so that it can be asserted with a given probability that the value of the estimated parameter is in this interval. In other words, the confidence interval with a certain probability contains the unknown value of the estimated quantity. The wider the interval is, the higher the inaccuracy is. We set the threshold to above 0.5 because it is assumed that the UAV operates in an untrusted and uncertain environment, where it can be affected by a large number of factors. At the same time, the UAV is able to cope with factors such as the wind and GPS inaccuracy and can operate in such conditions. We decided that the possibility of deviations of 10–15% were as acceptable ones. The thresholds were obtained experimentally. When we were calculating the confidence interval, we obtained a probability function of how far the current value of the parameter fell within the boundaries of the confidence interval. This value should tend to be 1. Accordingly, a number that is greater than 0.8 tends to be 1. We observed values from 0.6–0.9; they tended to be 1.

The result of calculating the cumulative Poisson function is shown in Figure 5. Figure 5 shows that the values for longitude and latitude almost always lie within the confidence interval, that is, they do not change dramatically. There are only a few outliers in the time interval for the longitude and latitude, but they are not significant, and there are no more than three such values in a row. This may be due to the UAV turning when it corrects the course. At the same time, when we were calculating the cumulative Poisson function for the case with a cyber-attack, a significant deviation was observed, for example, as shown in Figure 6.

In this scenario, the attack did not start immediately, but it occurred sometime into normal operation. This situation is shown in Figure 6. At the same time, the values of the cyber-physical parameters did not fall into the confidence interval because the UAV changed its altitude and flight speed quite sharply and received new coordinates. Having received fake coordinates, the UAV sought to move to a false waypoint, thinking that it was in the wrong place. That is why its flight speed increased. The calculation results are shown in Figure 7.

Of course, the UAV’s altitude may have fluctuated slightly during normal flight, but it remained mostly unchanged. The situation is the same with the flight speed, which is confirmed by Figure 8.

Figure 9 shows the mode resulting in confidence functions. In fact, for the case when an attack was being carried out, our algorithm would not allow it to reach the step of calculating the trust in the process, since for the case with an attack, the parameters did not fall into the confidence interval. Nevertheless, this calculation was made in the framework of our study.

Thus, we can see that all of the modes were verified successfully in the case of normal flight. In the event of an attack, the flight mode was not verified due to the failure of all of the indicators to fall within the boundaries of the confidence interval. In this situation, the system can determine that an attack is underway with a fake navigation signal or that a system fault has corrupted the navigation data. Either way, it can return the UAV to the takeoff point. The UAV can return to the takeoff point by disconnecting from the global navigation system, and then relying on its inertial navigation system, technical vision system, or built-in sensors.

Now, let us consider a situation where an attack on the UAV was not carried out, but the anomaly that arose nonetheless led to the UAV falling. It is necessary to assess whether our approach could detect such an anomaly if it were installed on a UAV. In this scenario, the UAV system crashed due to unforeseen reasons. This failure triggered the “Failsafe mode” when the vehicle encountered a problem during flight, such as loss of manual control, a critically low battery, or an internal error. Thus, we were dealing with an unforeseen situation that caused a system failure, the UAV deviated from the flight path, and as a result, the UAV crashed. Figure 10 shows two UAV flight options in the mission mode. In the first case, the UAV successfully completed the mission, flying through all of the established points (Figure 10a). In the second case, the UAV crashed due to unforeseen circumstances. Figure 10b shows that the UAV initially began to deviate from the course and could not even reach the first point. At the same time, no spoofing attacks were carried out, the UAV was recorded on 17 satellites, and the jamming and noise indicators were normal.

Let us analyze the results of calculations of Metric 1. The probability of it falling into the confidence interval for the estimated parameters is shown in Figure 11.

Figure 11 shows that the longitude under the stress condition initially did not fall into the confidence interval, and starting from the 31st time interval, the latitude did not fall into the confidence interval either. Initially, the drone may have been launched at the wrong point and shifted, but then it straightened the flight path, and at the moment starting from the 31st time interval, it finally moved away from the given trajectory. These calculations correlate with the real situation, as can be seen from Figure 11: the UAV first simply shifted from the green line, and then began to maneuver. The situation of a normal flight is highlighted in blue. Here, you can see that the coordinates completely match the expected ones. As for the flight speed, it also did not report the expected one. The advantage of monitoring the changes is that we already obtained a finite value of the probability of it falling into the confidence interval. In fact, we can evaluate each parameter using one formula and obtain uniform values that we can unambiguously attribute to being trusted or untrusted values. The flight altitude began to change in the area where the UAV flew with strong deviations from the route, and this was fixed using the first metric.

Let us consider the results of calculating Metric 2, which are presented in Figure 11. The mission mode is arranged in such a way that the UAV flew according to the flight task. At the same time, the speed and altitude of the flight specifically in this case should not have changed, so the rate of growth of the parameter should be unchanged. If the speed and altitude of the flight from point to point changed, then at certain moments, we would observe an increase in these parameters. In this experiment, the growth rate of the parameter should be constant, so the values should be 0.4–0.6. This parameter allows you to estimate how much the deviation from the expected value occurred, if there is a value of 0.5, then there is no deviation at all, since we allowed deviations of 10–15%, then the thresholds were selected based on this.

Figure 12 shows that the green graph, which indicates a normal flight, is almost always within the normal range, but some deviations are possible for the flight speed because for a small UAV, maintaining one speed is quite a challenge due to varying wind conditions. The rest of the parameters on the graphs are normal. As for a situation under the influence of stress, we observed deviations in all of the parameters. Deviations are associated with a sharp increase or decrease in the parameters, which are due to ongoing anomalous events with the UAV.

Let us consider the resulting trust value, which is the third metric shown in Figure 13.

It can be seen from Figure 13 that when an emergency occurred, the value of trust fell. The UAV did not immediately start to fall, but the confidence assessment allows the early detection of problems that have begun, giving it time to take corrective measures. In general, it can be said that as soon as the UAV began to deviate from the course, the level of confidence dropped significantly to zero. It would be possible to take control from the operator and correct the flight or call the UAV back. Nevertheless, this was not conducted in this specific case, and the UAV continued to fly further and crashed as a result of an unforeseen error.

To evaluate our approach, we used two metrics: anomaly (or attack) detection accuracy and the speed of anomaly detection. At the same time, when we were assessing the accuracy of detection, a Type I error and a Type II error were used. Methods for calculating these errors can be found elsewhere [34,35]. Table 4 presents the estimates of the anomaly and attack detection accuracy for the UAVs, as well as a Type 1 Error score for normal behavior scenarios.

When we were analyzing the results, two scenarios of anomalous behavior were considered. The first scenario is a GPS navigation signal spoofing attack. This attack is described in our previous paper [31]. The results of the attack are analyzed, and the final verification is presented in Figure 8.

The second anomaly presented in Figure 9b is a software failure. The failure was accompanied by an incorrect change in the flight mode and a drone crash. The software crash is caused by a problem with the companion’s computer sending the wrong command to the flight controller. This failure occurred due to the introduction of errors into the control program, which should have been activated after the timer was set, and caused incorrect behavior and the submission of unexpected commands to the drone, in particular, an unauthorized change in the flight mode. The architecture of the experimental stand has been presented previously [32].

The countermeasures for a GPS spoofing attack can be a transition to an inertial navigation and communication system. As can be seen from the analysis of the processes during the spoofing attack, the UAV did not fly as expected by the flight task, therefore, it was necessary to correct the flight trajectory. In the case of an anomaly associated with a software failure of the companion’s computer, it is advisable to switch to autopilot mode and lock the companion’s computer control interface. In practice, the flight plan can be recorded on the flight controller before the start of the flight, and the UAV can then safely continue its flight, avoiding a possible crash.

The third version of the impact scenario was to implement an attack involving jamming of the UAV control channel. The attack consisted of sending a signal with more power at the same frequency at which the drone was controlled by the operator. Since the test drone did not have a foreseen response to the situation, the drone crashed after losing contact with the operator. This problem could be solved by modifying the firmware to introduce an emergency response algorithm.

When we were evaluating Type 1 Errors during normal flights, threshold exceedances were sometimes observed, but such events were isolated and not synchronized for different parameters. As a result, Metric 3 includes the evaluations of Metrics 1 and 2 and the Mode Flight, i.e., 11 values in total. If half of the values have normal values that fall within the range of the threshold, then the accuracy of the estimate will be greater than 0.5. This is the minimum threshold that can be set when one is evaluating accuracy. In general, an attack (or anomaly) affects the system if at least three or four parameters do not fall within the required range of values. Then, the accuracy threshold can be increased to 0.7.

From the two columns in Table 3 where an anomaly or attack was assessed, it can be seen that Metric 1 and Metric 2 complement each other. For example, in the case of anomalous activity for longitude in Metric 1, the probability value is 0.27, that is, Metric 1 does not indicate the presence of an anomaly, and this means that the longitude changed within the confidence interval. At the same time, the longitude has a value of 0.96 in Metric 2, which means that the number of time intervals in which a sharp change in the value was observed is significant, and an anomaly was detected there. Therefore, we conclude that the UAV moved within the specified range, but rather intensively, since usually the longitude changes smoothly when the UAV performs a mission.

Now let us consider an example for the attack case. In Metric 1, the latitude has a detection accuracy of 0.71. This means that the latitude value did not fall within the confidence interval. At the same time, Metric 2 shows a low latitude detection accuracy value of 0.24, that is, the latitude changed smoothly, and no jumps or sharp changes were observed.

When we were assessing the anomaly case of all of the parameters, the signal-to-noise ratio parameter did not change, and the flight altitude fell within the confidence interval and did not change sharply. However, the final detection accuracy for most of the time intervals exceeded the specified threshold, and an anomaly can thus be detected at the first appearance of non-standard changes in the system. In the attack case, the flight mode was not changed, the signal-to-noise ratio did not show frequent abrupt changes, and the resulting accuracy exceeds the threshold. Thus, calculating only the probability of it falling into the confidence interval or estimates of the growth/fall of the parameter value does not give an estimate that is as accurate as the combination of metrics and the estimation of the final Metric 3.

Now, let us analyze the attack detection rate using the example in Figure 14.

Here, the beginning of a navigation signal spoof attack is characterized by a decrease in the number of satellites that are being tracked. A decrease in the number of satellites was observed in time interval 115 (Figure 14a). Each time interval has a duration of 0.2 s. Figure 13b shows that at time interval 114, the value of the final Metric 3 was 0.65, and at time interval 116, it decreased to 0.5. That is, if we consider a value that is below or equal to 0.5 as a threshold value, then we can say that detection occurs, and all of the subsequent values are even lower. So, in time interval 118, the value of Metric 3 is already 0.4, which indicates that more than half of the parameters began to change to a non-standard value or went beyond the boundaries of the confidence interval, and therefore, an attack was being performed (or some other anomaly has been observed). Thus, the detection of an attack and the decrease in the level of trust in the process occur in less than a second.

Such rapid detection allows timely responses to anomalous incidents to be performed. When an anomaly is detected, it is necessary to reduce the level of trust in the system and report an unstable state immediately. In our experiments, primary detection occurred in all of the attack and anomaly cases. At the same time, a percentage of false negatives can appear when the system is under the influence of an attack for a long period of time, and some system parameters have an opportunity to adapt to the changed conditions. For instance, in the case of an attack, the drone did not crash and still tried to fly along a given trajectory, and therefore, the latitude and longitude values periodically fell into the confidence interval. Nevertheless, the attack is still detectable, and during the attack, our method allowed us to identify the state of the system under the influence of stressful conditions.

## 4. Discussion and Conclusions

It is critical to provide trust and security for cyber-physical systems. This is because CPSs interact directly with the physical world, the environment, and humans. At the same time, component failures and information attacks can directly affect the end sensors, actuators, microcontrollers, etc. [36,37]. In addition, if a CPS operates in an untrusted environment, it may be affected by natural factors such as weather conditions. When one is forming a threat model, countering attacks, and assessing risks, first of all, it is necessary to draw a clear description of the system and evaluate all of the information flows, as well as system processes.

Thus, the state of the art of our study is as follows:-We developed a method for calculating trust levels, which allows the CPS to assess the trust in the processes that occur during processing or flying in the case of UAVs in an autonomous mode. Such a system can be scaled for a group of UAVs, as they can exchange their values and build the ratings and reputation of a neighbor.-Our developed method uses normalized parameter estimation. The assessment uses two metrics. One of the metrics allows us to determine whether the process goes beyond the established confidence interval, and the second one determines the growth rate of the parameter. The growth rate of the parameter is important for certain processes, such as starting or stopping, while when the system performs the same type of monotonous actions, the process must be static. The two metrics are combined to produce an overall level of confidence in the process, making it easier to decide whether to respond.-Our approach allowed us to evaluate the trust in the process, taking into account the operating mode of the system, and this is an important aspect because in each mode, the device can behave differently. In addition, we directly monitored the change in the modes and evaluated how much this change was trusted. Thus, considering trust in conjunction with the processes in the system and expressing trust not in the system, but in the process, allows us to recognize emerging malfunctions. This allows error checking functions, not to exclude the system from work, but to correct its behavior. Such an adjustment is very important for dynamic, autonomous systems such as UAVs.

We considered the trusted operation of cyber-physical processes based on an assessment of the system’s state and operating mode. To detect operating mode anomalies, we have developed a method for verifying the state of a CPS based on data analysis. To this end, the following has been conducted:-An analysis of the CPSs’ features, their controllers, and the data they transmit;-The development of a functional model of the CPS based on the analysis of the operating modes and processing states to improve the fault tolerance;-The development of a method for verifying the processing state of the CPS and assessing the possibility of transitioning to another state;-Experimental studies and confirmation of our methods, algorithms, and software services.

Here, an experimental study was carried out for all of the operating modes of an unmanned aerial device. The data available in the public domain made it possible to evaluate only a limited set of modes and the impacts on the UAV. However, an evaluation with the available data showed that the verification method works well, and the final value of the verification makes it possible to assess the presence of an anomaly with high accuracy. When we were evaluating the confidence intervals for the no-attack mode, all of the parameters passed the required threshold of 0.5 and met the expected performance standards. At the same time, due to the fact that the cyber-attack caused the UAV to deviate from the set target point, the values did not fall into the trusted interval during the attack.

Our new method allowed us to verify changes in the control processes from two sides. In the case of calculating the confidence interval, the probability is estimated that the process should not go beyond the allowed limits, that is, it operates within the established limits. If the process goes beyond these limits, then this can be assessed as an anomaly.

When the cumulative Poisson distribution is calculated, some outliers may present. UAVs, in particular, do not operate under ideal conditions. Often, wind, rain, and fog require the device to correct its flight path. Therefore, isolated groups of outliers in this case are possible because the UAV must change its behavior under unusual circumstances. In order not to consider these as an anomaly, only three consecutive values that do not fall within the confidence interval were taken into account. As the experimental study showed, single outliers did not affect the final value of confidence.

Our new method can be used to implement software solutions aimed at detecting attacks and making decisions about the presence of malfunctions in the operation of a CPS.

As shown by the experimental study, this method makes it possible to determine not only the case when an attack is carried out on the UAV, but also other unforeseen situations. The recognition of such an anomaly is possible due to the decrease in the confidence value to zero, as shown by both experiments. Although the value was not reduced to zero during the attack, it reached 0.1. This is due to the fact that as a result of the attack, the spoofed UAV simply smoothly deviated from the route, and as a result of a different unforeseen circumstance, it fell [38]. At the same time, the analysis of different groups of parameters can make it possible to determine what caused the failure. The advantage of our method is also that it allowed us to analyze the trust in the connection with the process or mode of operation of a cyber-physical system. This is important because a CPS can behave differently in different modes and different processes consume resources with different intensity; what is normal for one process will be an attack or anomaly for another process.

Thus, the obtained results can be used to implement software solutions aimed at detecting system faults and cyber-attacks and thus make decisions about the presence of malfunctions in the operation of a CPS. Importantly, the approach minimizes the need for the prior knowledge and initial data about the system to detect anomalies.

In further research, it has been planned that we will conduct experiments and obtain new flight data. In addition, the verification of other types of CPSs will be carried out, the impact of other factors on the change in cyber-physical parameters will be studied, and additional modes of operation of the CPS will be tested.

## Figures and Tables

**Figure 1 sensors-23-01996-f001:**
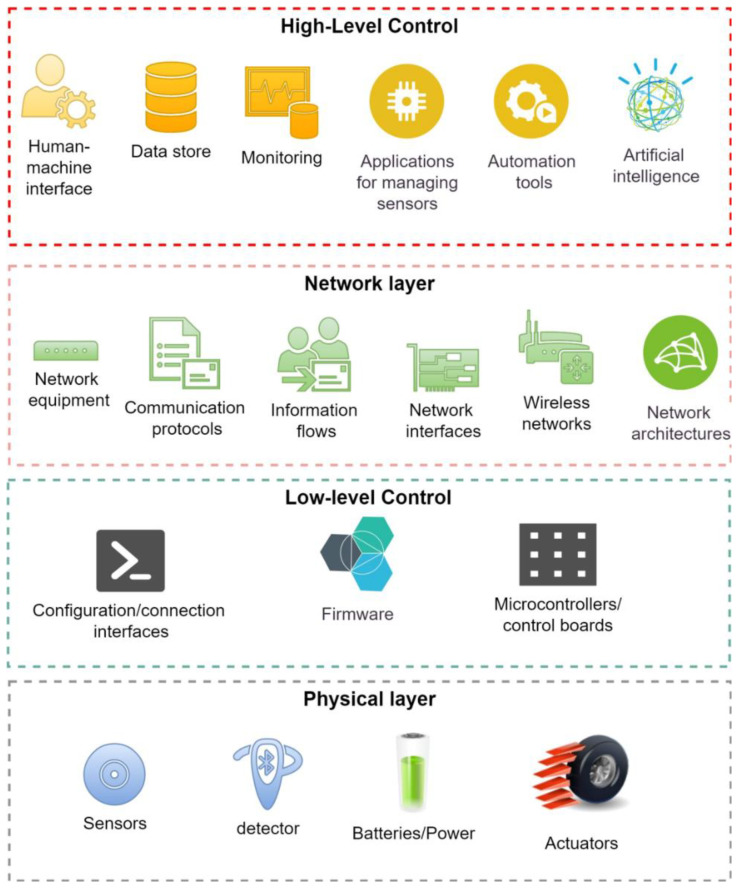
CPS architecture from the point of view of the functional model.

**Figure 2 sensors-23-01996-f002:**
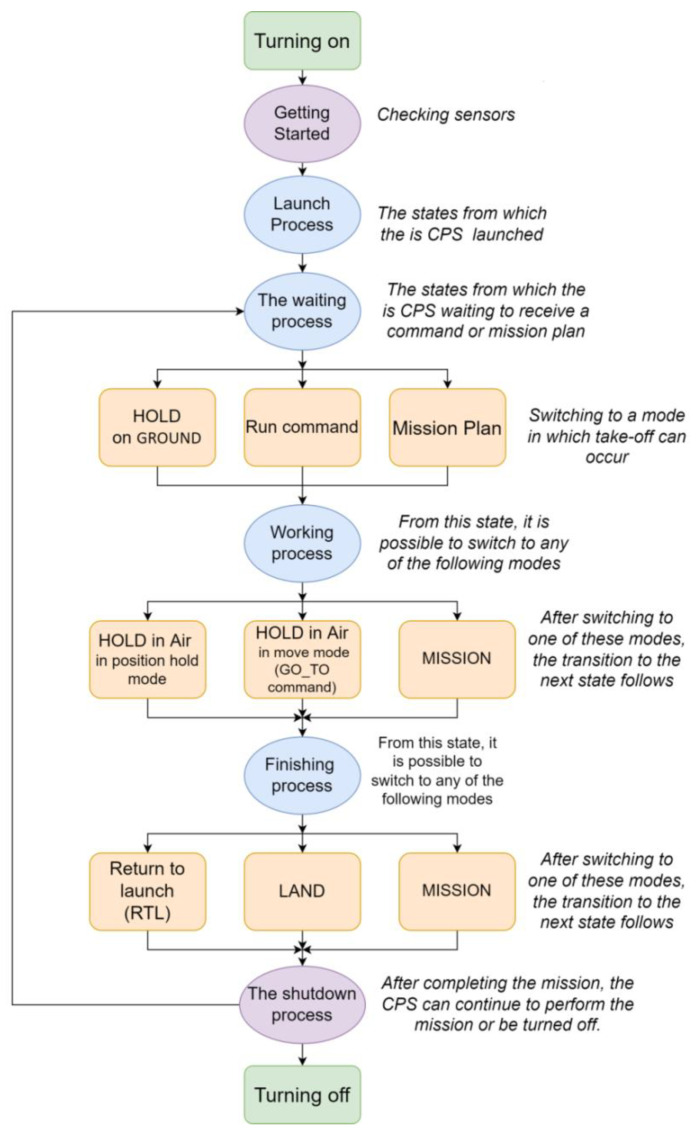
Scheme of possible transitions of processes through various modes of operation in the case of a UAV.

**Figure 3 sensors-23-01996-f003:**
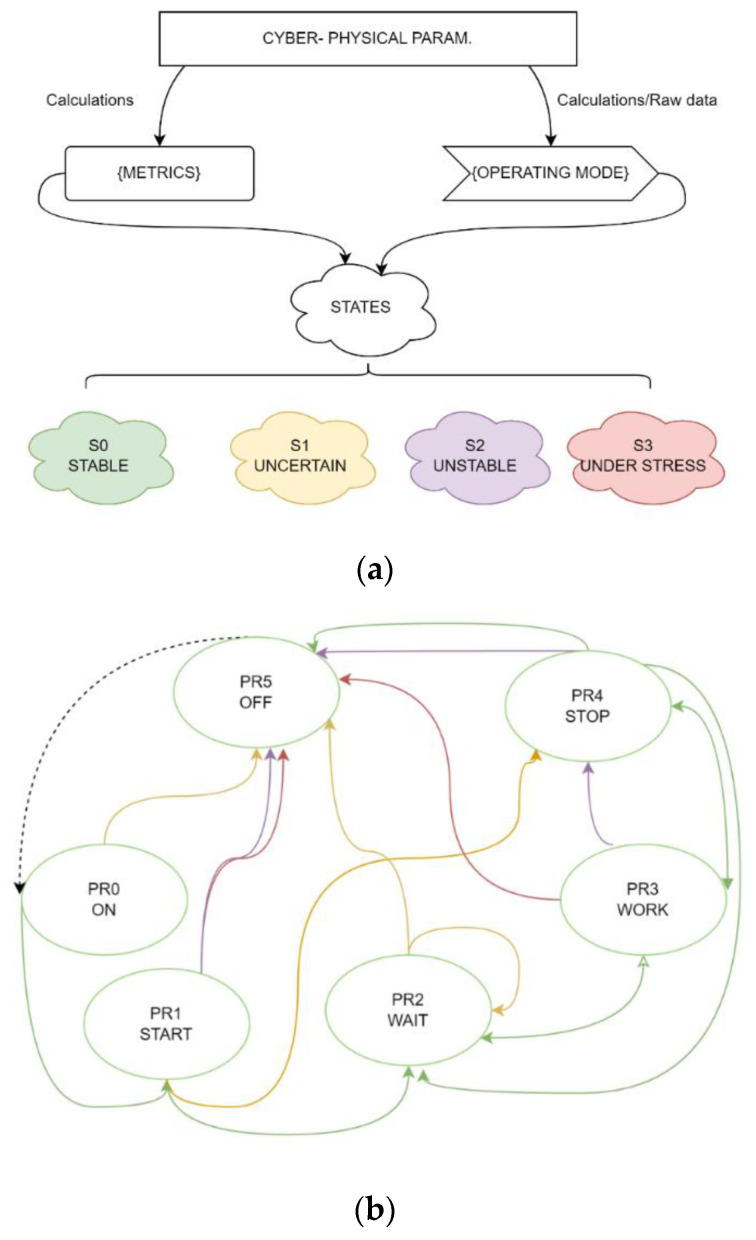
Scheme of conditions for transitions from one control process to another: (**a**) relationship between states and cyber-physical parameters, and (**b**) transitions from processes with the state’s consideration.

**Figure 4 sensors-23-01996-f004:**
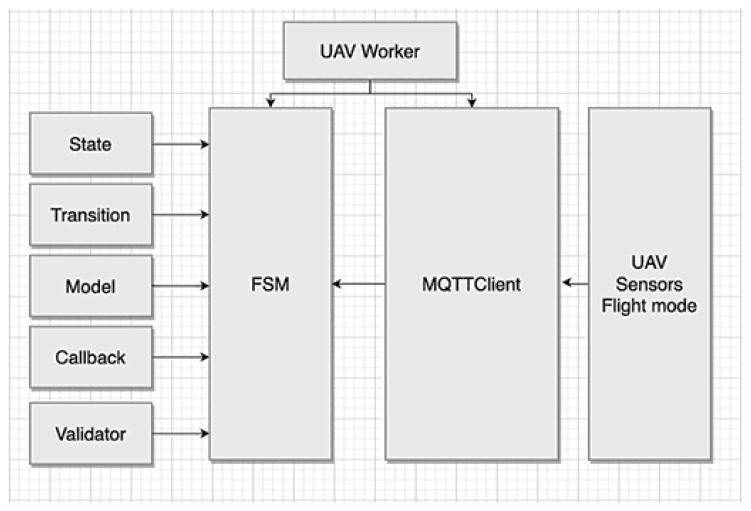
Structure of the software implementation.

**Figure 5 sensors-23-01996-f005:**
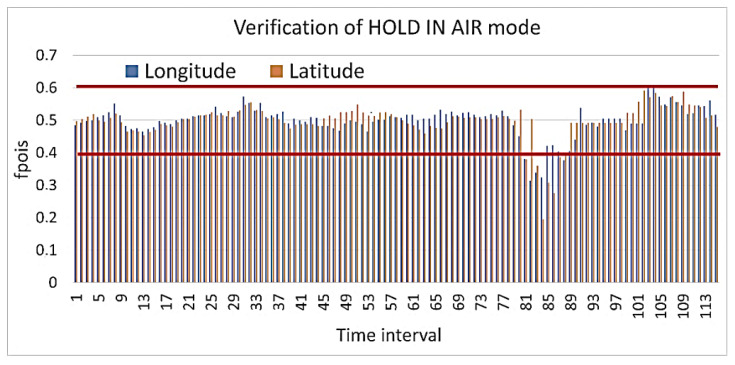
Verification of the HOLD IN AIR mode calculation of the cumulative function for the Poisson distribution.

**Figure 6 sensors-23-01996-f006:**
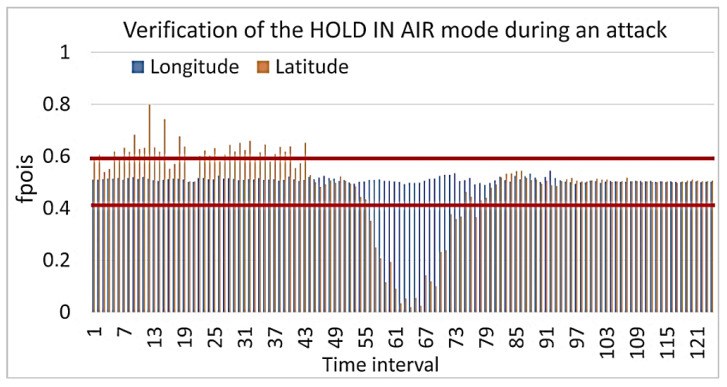
Verification of the HOLD IN AIR mode calculation of the cumulative function for the Poisson distribution during a cyber-attack.

**Figure 7 sensors-23-01996-f007:**
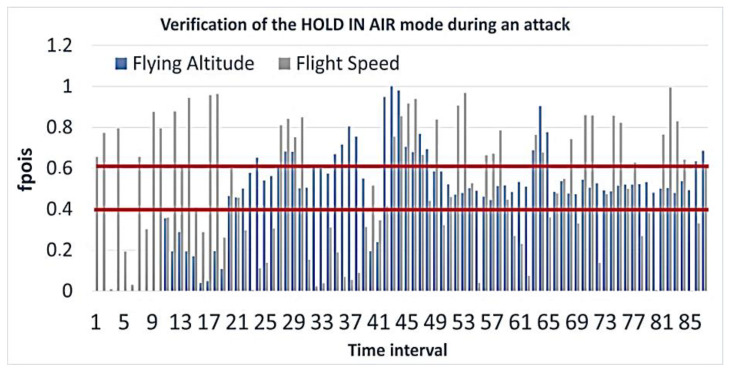
Verification of the HOLD IN AIR mode calculation of the cumulative function for the Poisson distribution during an attack.

**Figure 8 sensors-23-01996-f008:**
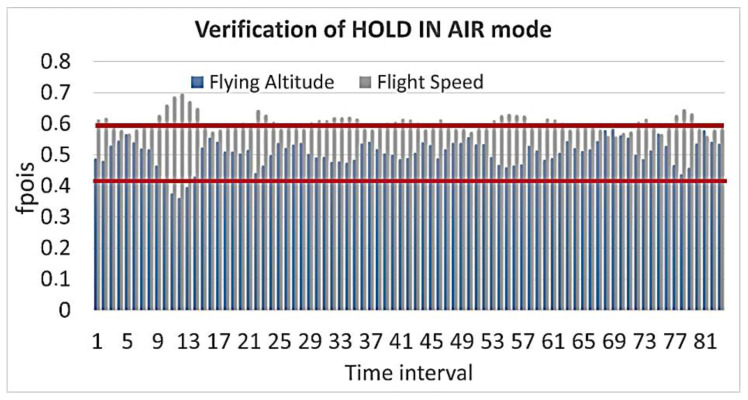
Verification of the HOLD IN AIR mode calculation of the cumulative function for the Poisson distribution.

**Figure 9 sensors-23-01996-f009:**
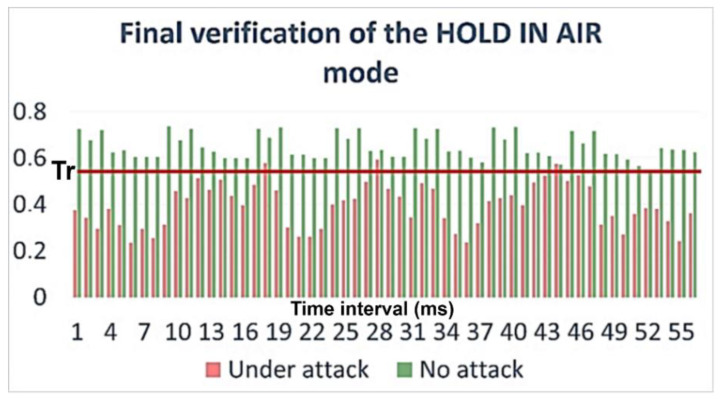
Verification of the HOLD IN AIR mode calculation of the trust function for the current process.

**Figure 10 sensors-23-01996-f010:**
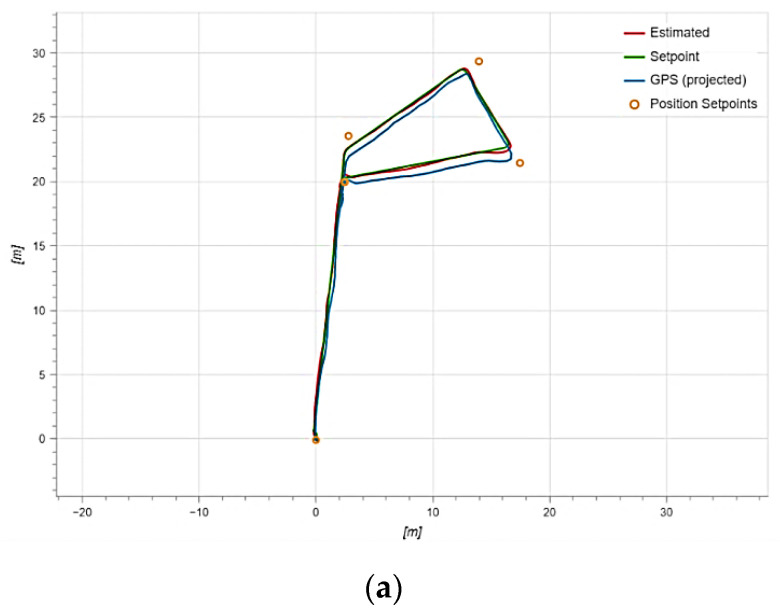
UAV flight trajectory during the mission: (**a**) under normal conditions and (**b**) under impact conditions.

**Figure 11 sensors-23-01996-f011:**
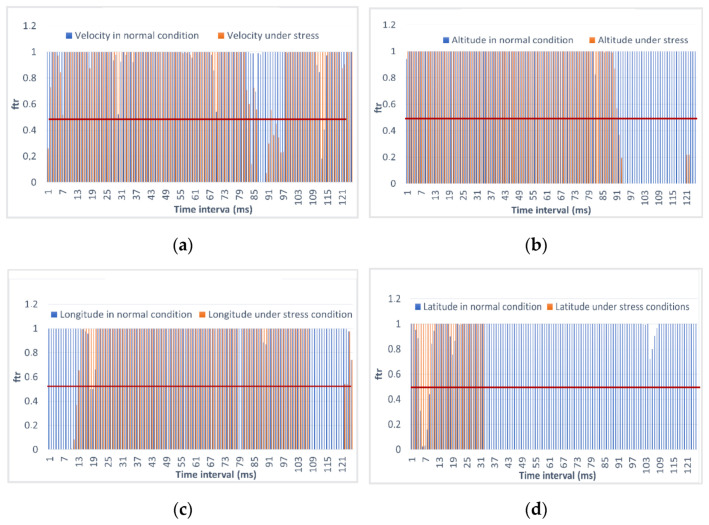
Evaluation of Metric 1 during flight in mission mode under normal and stress conditions for indicators of (**a**) flight speed; (**b**) flight altitude; (**c**) flight latitude; (**d**) flight longitude.

**Figure 12 sensors-23-01996-f012:**
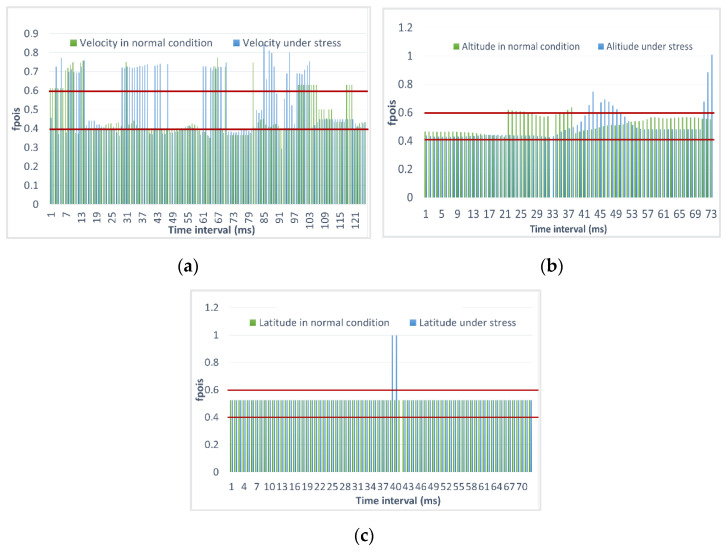
Evaluation of Metric 2 during flight in mission mode under normal and stress conditions for (**a**) flight speed; (**b**) flight altitude; (**c**) flight latitude.

**Figure 13 sensors-23-01996-f013:**
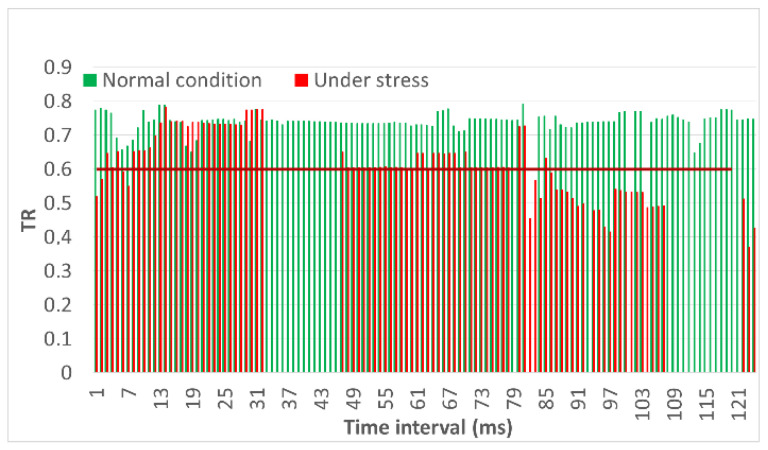
Evaluation of Metric 3 in flight in mission mode under normal conditions and under stress conditions for indicators.

**Figure 14 sensors-23-01996-f014:**
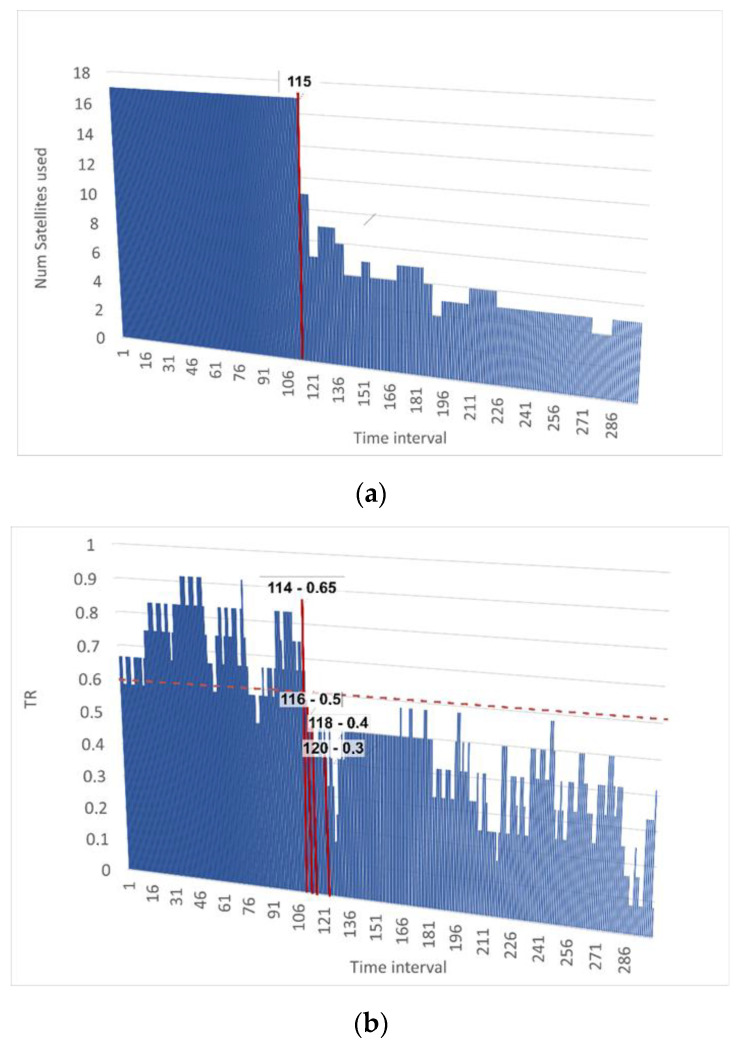
Estimating time of attack detection using state detection: (**a**) number of satellites used and (**b**) level of trust (TR) in the process being executed.

**Table 1 sensors-23-01996-t001:** Comparison of UAV processes and modes.

Process	Mode
Getting Started $PR_{{}_{0,on}}^{}$	Cyber-physical parameters change values from False to True.
Launch $PR_{{}_{1,start}}^{}$	Mode MANUAL $om_{1,man}$,
	Mode TAKEOFF $om_{1,t}$.
Waiting $PR_{{}_{2,wait}}^{}$	Mode HOLD ON GROUND $om_{1}$,
	Mode MISSION $om_{2,m}$.
Working $PR_{{}_{3,proc}}^{}$	Mode MISSION $om_{2,m}$,
	Mode HOLD with movement $om_{2,h}$.
Finishing $PR_{{}_{4,stop}}^{}$	Mode LAND $om_{3,l}$,
	Mode Return to launch (RTL) $om_{3,rtl}$.
Shutdown $PR_{{}_{5,off}}^{}$	Mode Shutdown $om_{4,off}$,
	Cyber-physical parameters $CP_{gc}$, $CP_{ac}$, $CP_{mc}$, $CP_{lp}$, $CP_{gp}$, $CP_{hp}$
	change values from True to False.

**Table 2 sensors-23-01996-t002:** A set of metrics for verifying the state of the system’s currently executing process.

Metric	Equations	Description
1. Reliability of performed functions in the current state		
1.1. Confidence interval limits in the presence of target indicators		
1.1.1. Lower limit of confidence interval $CP_{i}^{\min}$	$CP_{i,st}^{\min}=CP_{i,st}-\sigma^{st}{}_{i}$ $CP_{i,s_{n-1}}^{\min}=CP_{S_{n}-1}-\sigma_{i}$ $CP_{i,current}^{\min}=\bar{CP_{i}}-t\cdot \sigma^{current}{}_{omi}/\sqrt{n}$	Term $CP_{i,st}^{\min}$ is the minimum value in the presence of target indicators, $CP_{S_{n}-1}$ is the value of the cyber-physical parameter from the previous state, $t*\sigma /\sqrt{n}$ is the accuracy of estimate, *t* is the argument of the Laplace function, where $\Phi (t)=\frac{\alpha }{2}$, *α* is the given reliability, $\sigma_{i}$ is the allowable deviation, and $CP_{i,current}^{\min}$ is the minimum value based on collected parameters for previous time intervals.
1.1.2. Upper limit of the confidence interval $CP_{i}^{\max}$	$CP_{i,st}^{\max}=CP_{i,st}+\sigma^{st}{}_{i}$ $CP_{i,s_{n-1}}^{\max}=CP_{S_{n}-1}+\sigma_{i}$ $CP_{i,current}^{\max}=\bar{CP_{i}}+t\cdot \sigma^{current}{}_{omi}/\sqrt{n}$	Term $CP_{i,st}^{\max}$ is the maximum value in the presence of the target indicators, $CP_{S_{n}-1}$ is the value of the cyber-physical parameter from the previous state, $t*\sigma /\sqrt{n}$ is the accuracy of estimate, *t* is the argument of the Laplace function, where $\Phi (t)=\frac{\alpha }{2}$, *α* is the given reliability, $\sigma_{i}$ is the allowable deviation, and $CP_{i,current}^{\max}$ is the maximum value based on collected parameters for previous time intervals.
1.1.3. Allowable deviation $\sigma_{i}$	$\sigma_{omi}^{st}(CP_{i})=\sqrt{\frac{\sum (CP_{i}\;-\;\bar{CP_{i}}{)}^{2}}{n}}$ $\sigma_{omi}^{current}(CP_{i}\Delta w)=\sqrt{\frac{\sum (CP_{i}\Delta w\;-\;\bar{CP_{i}}\Delta w{)}^{2}}{n}}$ $\sigma_{i}(CP_{i})=const$	Function $\sigma_{omi}^{st}$ is the standard deviation obtained during the reference cycle of the CPS for a given mode, $\sigma_{omi}^{current}$ is the standard deviation obtained during the current operation of the CPS based on the collected parameters for previous time intervals, $\bar{CP_{i}}$ is the average parameter value, $CP_{i}\Delta w$ is the parameter values for the previous interval of CPS operation, *const* is a constant because, in some cases, the allowable deviation can be set by a constant if anomaly indicators have been established by the system documentation.
1.2. Probability of falling within the confidence interval		
1.2.1. Function $f_{tr}$ of the probability of falling into the confidence interval for a cyber-physical parameter	$\begin{array}{l} f_{tr}(CP_{i}^{\min}<CP_{i}<CP_{i}^{\max})=\\ =\Phi \left(\frac{CP_{i}^{\max}\;-\;CP_{i}}{\sigma_{i}}\right)\;-\;\Phi \left(\frac{CP_{i}^{\min}\;-\;CP_{i})}{\sigma_{i}}\right) \end{array}$	This function allows us to check the compliance with the process parameters with the maximum permissible values.
2. Control of increase/decrease in cyber-physical parameters		
2.1. Cumulative function for the Poisson distribution $f_{pois}$	$f_{pois}(CP_{i}|\;\bar{CP_{i}})=\sum_{j=1}^{CP^{n}{}_{i}}\frac{{\bar{CP_{i}}}^{CP_{i}}e^{-\bar{CP_{i}}}}{CP_{i}!}$ $f_{pois,\min}(CP_{i}|\;CP_{i,\min})=\sum_{j=1}^{CP^{n}{}_{i}}\frac{CP_{i,\min}{}^{CP_{i}}e^{-CP_{i,\min}}}{CP_{i}!}$ $f_{pois,\max}(CP_{i}|\;CP_{i,\max})=\sum_{j=1}^{CP^{n}{}_{i}}\frac{CP_{i,\max}{}^{CP_{i}}e^{-CP_{i,\max}}}{CP_{i}!}$	The cumulative Poisson probability is related to the probability that the random Poisson frequency is more dependent on a given limit and less dependent on a given upper limit. Term $CP_{i,\max}$ is the upper limit of the value of the cyber-physical parameter, and $CP_{i,\min}$ is the lower limit of the cyber-physical parameter.
2.2. The average value of the cyber-physical parameter in the range of the sliding window $\bar{CP_{i}}$	$\bar{CP_{i}}=\frac{1}{n}\sum_{j=1}^{n}CP_{i}\Delta w_{ij}$	Term *n* is the sample size, $CP_{i}$ is the sample parameter values, and Δw is the sliding window for a given time interval of values, which is equal to *n*.
2.3. Process change trust function $f_{pois,tr}$	$f_{pois,tr}(CP_{i,\max}|\;CP_{i,\min})=\frac{f_{pois,\max}}{f_{pois,\min}}$	This function determines the probability of the current value of the cyber-physical parameter approaching the target value.
3. Trust in the current process		
3.1. Process trust function	$\begin{array}{l} Tr(f_{tr}(\{CP_{i}\}),f(om_{i})=\\ =\frac{{\sum f(\{CP}^{i}{}_{ph}\})}{N}\times f(om_{i}) \end{array}$	This function is determined based on the confidence in the parameters, as well as on the basis of the value from the intrusion detection system and the corresponding mode. Function $f(om_{i})$ determines compliance with the mode; it can take the values of 0 or 1.

**Table 3 sensors-23-01996-t003:** Data transmitted from the device in real time.

Topic	Description
System/controller/out/info/mode	Flight Mode
System/controller/out/info/GPS	Information about GPS
System/controller/out/info/telemetry	Telemetry
System/controller/out/info/position	Positioning
System/controller/out/info/odometry	Odometry
System/controller/out/info/status/flight controller	Drone Controller Status
System/controller/out/info/status/battery	Battery Information
System/controller/out/info/status/mission	Mission Progress
System/controller/out/info/status/sensor	Sensor Status

**Table 4 sensors-23-01996-t004:** Detection accuracy estimates.

Parameter	Normal Flight 1 (Type I Error)	Normal Flight 2 (Type I Error)	Anomalous Flight 1 (Software Failure)	Under Attack Flight 2 (GPS Spoofing)	Under Attack Flight 3 (Jamming the Control Channel)
Metric 1					
Latitude	0.001	0.015	0.93	0.71	0.6
Longitude	0.019	0	0.27	0.86	0.7
Speed	0.02	0.4	0.71	0.71	0.99
Altitude	0.04	0.03	0.35	0.87	0.97
Signal-to-Noise Ratio	0	0	0	1	1
Metric 2					
Latitude	0.08	0.08	0.14	0.24	0.5
Longitude	0.02	0.07	0.96	0.67	0.7
Speed	0.015	0.3	0.82	0.98	0.99
Altitude	0.05	0.04	0.125	0.76	1
Signal-to-Noise Ratio	0	0	0	0.24	1
Flight Mode					
Flight Mode	0	0	1	0	0
Metric 3					
Metric 3	0.001	0.02	0.89	0.97	0.98

## Data Availability

Data sharing not applicable.

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
