# Peer review of "Trusted Operation of Cyber-Physical Processes Based on Assessment of the System’s State and Operating Mode"

_sensors, 2023, doi:10.3390/s23041996_

Round 1

Reviewer 1 Report (New Reviewer)

Overall

Paper has little technical contents but full with related works and results. Especially, the introduction is only with related works followed by one paragraph of authors' own technology lacking the clear description.
This makes the readers/reviewers difficult to judge the contribution. I think authors need to reorganize the technical contents in introduction, methodology completely.

Strong points

Created a new method that uses system state to determine if the current system operation is in the trusted zone.

Extensive related works (Too long)

Weak points

There is no explanation as to what kind of cyber-attack came in. I would like to see more verifications to see if appropriate countermeasures are possible for various attack situations.

At Fig 5, it seems that checking is possible only after some time has passed when the cyber-attack started. However, if the attack is not quickly dealt with after it has been launched, it can lead to fatal results.

In the equation of 2.2, the performance of the system is expected to vary by ∆w. I would like you to share the results of variables before Result to see what effect it has.

Additional comment

*Introduction is too long

*P13 CP^max (i,st) = CP, ist -

*Please add some full word with unit at graph axis in Result part

*P19 - so the values should be from 0.4–06. -> 0.4-0.6

*Fig 10 -> It would be better to write only one subtitle at the top center or delete it.

Author Response

Dear Reviewer,

Thank you very much for reviewing of our paper, and for your comments and detailed suggestions helped us to improve our paper. We hope that we have understood them well. In accordance with your comments and suggestions as well as suggestions of other reviewers, we have improved our manuscript as much as possible within the allotted time. All the changes/improvements are also highlighted in the Highlighted Changes file.

Reviewer 1:

Overall

Paper has little technical contents but full with related works and results. Especially, the introduction is only with related works followed by one paragraph of authors' own technology lacking the clear description.

This makes the readers/reviewers difficult to judge the contribution. I think authors need to reorganize the technical contents in introduction, methodology completely.

Strong points

Created a new method that uses system state to determine if the current system operation is in the trusted zone.

Extensive related works (Too long)

Response to Reviewer 1:

Previously, we had to extend the description of related works in accordance with the suggestions of other reviewers given at the previous evaluation stage of the paper. We have reorganized the introduction. In particular, we tried to shorten the description of the considered works as much as possible and focus on our development. In addition to shortening the text, we added the following sentences in Introduction.

On page 4:

“However, their system may be ineffective since the base station can also be attacked and is a single point of failure. In addition, the CPS node may be out of the field of view of the base station. Our approach, in contrast to this study, allows the CPS node to independently make decisions about the trust in control processes.”;

“In contrast to their study, our approach allows the node to act autonomously and correlate its expected behavior with the behavior of the group on its own, which increases the degree of autonomy and hence reliability”

On page 5:

“In contrast to their method, in our approach the node relies on the readings of internal sensors and its estimates can be built on the basis of a combination of readings from any sensors at the physical or firmware level.”;

“However, use of cryptographic techniques to validate trust may not be appropriate for a CPS. This is because an attacker can take over control of the node. That is why the node itself must analyze the changes in processes and evaluate its trust in them. If a node is captured by an intruder, it must determine this itself and implement countermeasures, as is possible in our approach.”

“By contrast, however, the advantage of our approach is that knowledge of the normal values can simplify calculations but is not required. In addition, our method evaluates the dynamics of the process, and controls the behavior of the process in different modes of operation, which increases the accuracy of the assessment when determining anomalous behavior.”;

“However, a disadvantage of their approach is that the event graph can be quite large and indicates that before the system was launched, a detailed analysis of it and full control of all possible transitions must be carried out. Such a system cannot be applied to different types of UAVs and cyber-physical systems. In contrast to their approach, we rely on process control at a higher level and describe the process by changes in cyber-physical parameters, which makes it possible to carry out analysis by mathematical methods.”;

“This work is aimed at assessing anomalies in the operation of mechanical systems. This work has similarities with ours, since the mechanical process that the authors consider is similar to cyber-physical processes. However, the authors write that their process requires a large amount of data and requires precise thresholding to reduce the number of false positives. For our approach, the thresholds are determined based on methods of probability theory and do not require a long adjustment.”.

We also expanded the Materials and Methods section, where we outlined the implementation of the method and the algorithm:

“2.3 Development of a software module for a state analysis machine

Our implementation of the experimental software module uses the Python programming language Version 3.8. To transfer data from sensors and transmitters, a message broker of the MQTT (Message Queuing Telemetry Transport) protocol was used. The developed architecture uses the event-driven approach (event-driven architecture).

The event-driven architecture pattern is a popular distributed asynchronous architecture pattern used to build applications. This is a modern approach to design based on data describing “events”. An event-driven architecture allows an application to respond to these events as they occur. The main benefit of an event-driven architecture is responsiveness. Since everything happens in real time, this architecture provides the fastest response time.

The event-based approach has become extremely popular in recent years due to both the great growth of data sources that generate events (IoT sensors) and the development and adoption of technologies that process the flow of events, such as Hazelcast Jet and Apache Kafka.

An event in such a system is a message that represents data or commands when an observed value changes, such as an increase or decrease in the flight vehicle altitude. This message or event is generated by what is called an event emitter, which is an entity that detects a change and notifies the system.

In our prototype, the sensors of the flight vehicle are the event generator. A common design pattern that is used to implement this process is the publish/subscribe pattern. In this case, the event emitter is called the publisher, and the stakeholders are called event subscribers or event handlers.

MQTT is one of the most widely adopted IoT communication protocols that supports the above architecture and is based on a publish/subscribe communication pattern using an MQTT broker to coordinate event delivery. The broker will receive events from the producer and forward them to subscribers.

As part of the development, a specification for data transfer via the MQTT protocol was designed. Table 3 lists the main topics for getting data from a flight controller. The main topics for getting data from a flight controller are listed in Table 3.

Table 3. Data transmitted from the device in real time.

Topic

Description

System/controller/out/info/mode

Flight Mode

System/controller/out/info/GPS

Information about GPS

System/controller/out/info/telemetry

Telemetry

System/controller/out/info/position

Positioning

System/controller/out/info/odometry

Odometry

System/controller/out/info/status/flight controller

Drone Controller Status

System/controller/out/info/status/battery

Battery Information

System/controller/out/info/status/mission

Mission Progress

System/controller/out/info/status/sensor

Sensor Status

The general structure of the data acquisition methodology is shown in Figure 4.

Responsibilities of the elements are as following:

  • UAV Worker – The main class for launching the creation and initialization of all objects of other modules;
  • FSM (Final State Machine) – Responsible for the context in which the UAV operates, stores the state of the cyber-physical system, and meets the conditions for transitions between states;
  • State – The base class for describing the state;
  • Transition – The base class for transitioning and calling checks for conditions of transitions between states;
  • Model – The base class of the context in which the UAV operates;
  • Callback – The base class for how callback methods work;
  • Validator – The base class of the transition condition; and
  • MQTT Client – The module responsible for receiving data from sensors, and transmitting control commands to the flight controller.

Figure 4. Structure of software implementation.

2.4 Algorithm of the State Machine for determining the state of the UAV

Our state machine for a UAV aims to track whether the process under observation goes outside the confidence interval. In addition, it must assess whether the process is changing according to expectations. For this, the metrics presented above were used. Estimation of the probability of values falling within the boundaries of the confidence interval is made if the parameter exceeds a threshold of 0.5 because the parameter’s value varies from 0 to 1; the closer to 1, the greater the probability of falling into the confidence interval. Given the allowable deviations, a value above 0.5 is acceptable; this threshold is set when creating a software module and is unchanged.

Next, it is important to evaluate the growth or fall of the parameter values. For the takeoff process, it is important that the acceleration and altitude (and in some cases also latitude and longitude) increase. The growth of the value is determined by the Cumulative Poisson function. It is designed in such a way that when there is an increase in the value, then there is a deviation from the average value upwards, and thus the probability that the function reaches the “desired” value (that is, what is above the average close to one) increases. Therefore, the growth of parameters can be determined by increasing the Cumulative function above a threshold of 0.6. If there is a drop in the value, and it becomes less than the average (or expected value), then the probability of reaching the “desired” value decreases, the value of the function tends to zero, and it will become less than 0.4. This situation is normal when the aircraft is landing, or the system is turned off. These thresholds are also embedded in the software module. The software module works according to the following algorithm.

  1. Determine the current flight mode.
    • If the UAV is in the current flight mode and switches to a new flight mode, then determine whether this transition is legitimate according to the transition graph, as shown in Figures 2 and 3.
    • If the mode is legitimate and the transition to the mode is legitimate, then , if not, then it becomes 0.
  2. Determine the reliability of the functions performed in the current state.
    • From the flight task file it is necessary to take information about: altitude, flight speed, longitude and latitude, their maximum, and minimum values and waypoints.
    • Calculate the boundaries of the confidence interval using formulas from lines 1.1.1 and 1.1.2 in Table 2.
    • From line 1.2.1 in Table 2, calculate the function of the probability of falling into the confidence interval for each of the parameters.
    • If condition is met, then set the parameter to 1 and go to Step 3.
    • If condition is not met, determine the parameter for which this condition was not met and assign the value 0 to the parameter. Return the value of the parameter and go to Step 3 .
  3. Control the increase/decrease of the cyber-physical parameter (Metric 2).
    • Calculate the average value of the cyber-physical parameter for intervals using the formula from line 2.2 in Table 2.
    • Calculate the value of the cumulative Poisson function for each cyber-physical parameter using the formula from line 2.1 in Table 2.

3.2.1 If the current processes are , and , then , or  and goes to zero, then set the parameter to 1 and go to Step 4.

If the condition above is not met within  intervals, then set the parameter to 0 and return the parameter value, for which the condition is not met ( set is optional, default is 3), and go to Step 4.

If the current modes are = , , , , and , then set the parameter to 1 and go to Step 4.

If the condition above is not met within  intervals, then set the parameter to 0 and return the parameter value, for which the condition is not met ( set is optional, default is 3), and go to Step 4.

3.2.2 If the current processes are , and , then the processes change the confidence function (formula from line 2.3 in Table 2) for latitude and longitude calculated relative to the starting point. That is, it is necessary to take the coordinates of the point from which the UAV starts taking off, obtain the lower and upper bounds of the confidence interval, and then calculate the value of the function.

If the value for the parameters of altitude and velocity is > 0.6 and tends to 1, then assign a value of 1 and go to Step 4.

If  < 0.6 during the intervals , then assign a value of 0 and return the value of the parameter for which the condition was not met (set is optional, default is 3), and go to Step 4.

3.2.3 If the current processes are , and , then the process change confidence function ( formula from line 2.3 in Table 2) for latitude and longitude calculated based on the point to which the UAV is aiming (also taken from the flight task).

If the value for the parameters of altitude and velocity is < 0,4 and tends to zero, then assign a value of 1 and go to Step 4.

If  > 0.4 during the intervals , then assign a value of 0 and return the value of the parameter for which the condition was not met (set is optionally, default is 3), and go to Step 4.

  1. Calculation of the final value (Metric 3) of trust by Formula 3.1 in Table 2. (For the final trust value, the values of each metric for each parameter are taken, depending on whether the values calculated above tended towards 0 or 1.)
    1. If = 0, it is necessary to provide an emergency response.
    2. If > threshold value, it means that the state is stable. (The threshold value depends on how many metrics we allow to be zero. We will assume that if three or more metrics are equal to zero, then an anomaly occurs, so the threshold will be taken as 0.6. This value is optional and can be changed by the user when testing the algorithm.)
  • If = 0.6, the system is in an undefined state.
  1. If < 0.6, the systems is in an unstable state.

We used a floating window equal to three time intervals. If the size of the floating window is increased, then the speed of anomaly detection will decrease, if the size is decreased, then the accuracy will be reduced. This value can be changed depending on how often the values are received from the flight controller. In our case, the values are obtained three times per second. Based on this, a floating window was also defined. As the frequency of data collection increases, the window may change. These calculations do not affect system performance by themselves due to the presented architecture of the software application.”.

Reviewer 1:

Weak points

There is no explanation as to what kind of cyber-attack came in. I would like to see more verifications to see if appropriate countermeasures are possible for various attack situations.

Response to Reviewer 1:

We have added the following paragraphs after Table 4:

“When analyzing the results, two scenarios of anomalous behavior were considered. The first scenario is a GPS navigation signal spoofing attack. This attack is described in our previous paper [31]. The results of the attack are analyzed, and the final verification is presented in Figure 8.

The second anomaly presented in Figure 9b is a software failure. The failure was accompanied by an incorrect change of flight mode and a drone crash. The software crash is caused by a problem with the companion’s computer sending the wrong command to the flight controller. This failure occurred due to the introduction of errors into the control program, which should have been activated after the set timer and cause incorrect behavior, and the submission of unexpected commands to the drone, in particular, an unauthorized change in flight mode. The architecture of the experimental stand was presented previously [32].

Countermeasures for a GPS spoofing attack can be a transition to an inertial navigation and communication system. As can be seen from the analysis of the processes during the spoofing attack, the UAV does not fly as expected by the flight task; therefore, it is necessary to correct the flight trajectory. In the case of an anomaly associated with a software failure of the companion’s computer, it is advisable to switch to autopilot mode and lock the companion’s computer control interface. In practice the flight plan can be recorded on the flight controller before the start of the flight, and the UAV can then safely continue its flight, avoiding a possible crash.

The third version of the impact scenario was to implement an attack on the jamming of the UAV control channel. The attack consisted in giving a signal with more power at the same frequency at which the drone was controlled by the operator. Since the test drone did not have a foreseen response to the situation, the drone crashed after losing contact with the operator. This problem could be solved by modifying the firmware, to introduce an emergency response algorithm.

Table 4. Detection accuracy estimates.

Parameter

Normal

Flight 1

(Type I Error)

Normal Flight 2

(Type I Error)

Anomalous

Flight 1

(Software Failure)

Under Attack

Flight 2

(GPS-Spoofing)

Under Attack

(Jamming the Control Channel)

Metric 1

Latitude

0.001

0.015

0.93

0.71

0.6

Longitude

0.019

0

0.27

0.86

0.7

Speed

0.02

0.4

0.71

0.71

0.99

Altitude

0.04

0.03

0.35

0.87

0.97

Signal-to-Noise Ratio

0

0

0

1

1

Metric 2

Latitude

0.08

0.08

0.14

0.24

0.5

Longitude

0.02

0.07

0.96

0.67

0.7

Speed

0.015

0.3

0.82

0.98

0.99

Altitude

0.05

0.04

0.125

0.76

1

Signal-to-Noise Ratio

0

0

0

0.24

1

Flight Mode

Flight Mode

0

0

1

0

0

Metric 3

Metric 3

0.001

0.02

0.89

0.97

0.98

Reviewer 1:

At Fig 5, it seems that checking is possible only after some time has passed when the cyber-attack started. However, if the attack is not quickly dealt with after it has been launched, it can lead to fatal results.

Response to Reviewer 1:

Figure 5 shows the assessment of only one metric. We focus on the fact that only the totality of the metric gives a clear, accurate, and fast result. Figure 13 shows that during the final verification of the process, detection occurs quickly and takes tenths of a second, literally 0.4-0.6 seconds. In addition, the attack does not start immediately, but after some time after normal operation. An estimate of the attack detection time is shown in Figure 14.

Reviewer 1:

In the equation of 2.2, the performance of the system is expected to vary by ∆w. I would like you to share the results of variables before Result to see what effect it has.

Response to Reviewer 1:

We used a floating window equal to three-time intervals. If the size of the floating window is increased, then the speed of anomaly detection will decrease, if it is decreased, then the accuracy will be reduced. This value can be changed depending on how often the values are received from the flight controller. In our case, the values are obtained three times per second. Based on this, a floating window was also defined. As the frequency of data collection increases, the window may change. These calculations do not affect system performance by themselves due to the presented architecture of the software application.

Reviewer 1:

Additional comment

*Introduction is too long

Response to Reviewer 1:

We have to extend section Introduction in accordance with the suggestions of other reviewers given at the previous evaluation stage of the paper. So, we ask you do not insist on its reduction.

Reviewer 1:

*P13 CP^max (i,st) = CP, ist -

Response to Reviewer 1:

We have corrected it in Table 2.

Reviewer 1:

*Please add some full word with unit at graph axis in Result part

*Fig 10 -> It would be better to write only one subtitle at the top center or delete it.

Response to Reviewer 1:

The names of the axes and dimensions have been added to the charts. Also, the titles of the graphs were excluded in Figures 11, 12, 13.

Reviewer 1:

*P19 - so the values should be from 0.4–06. -> 0.4-0.6

Response to Reviewer 1:

It has been corrected for “0.4–0.6”.

Reviewer 2 Report (New Reviewer)

1. What is the main question addressed by the research? In the reviewing paper, the authors have considered the trusted operation of cyber-physical processes based on assessment of the system’s state and operating mode, and presented a method for detecting anomalies in the behavior of a cyber-physical system based on analysis of the data transmitted by its sensory subsystem.     2. Do you consider the topic original or relevant in the field? Does it address a specific gap in the field?   I consider the topic original and relevant in the field. It addresses the solution of a very important problem of increasing the level of trust in the operating modes and processes of cyber-physical systems. Cyber-physical systems include a wide range of devices, ranging from sensors to artificial intelligence. The cyber-physical system combines various elements. Ensuring the security of such systems requires new solutions.     3. What does it add to the subject area compared with other published material?   The authors have proposed and investigated a method for detecting anomalies in the behavior of a cyber-physical system based on analysis of the data transmitted by its sensory subsystem. Their method allows quick detecting of failures with high accuracy. Compared to existing methods, the authors applied several metrics at once to assess the degree of trust in the process. They evaluated the process based on the analysis of changes in cyber-physical parameters. After determining the specific parameters, the authors determine whether the change in the parameter goes beyond the boundaries of the confidence interval. Moreover, the authors define several options for calculating the boundaries. Next, the authors determine whether an increase in the parameter is observed. Their method is quite flexible and takes into account the dynamics of the system.     4. What specific improvements should the authors consider regarding the methodology? What further controls should be considered?   There is no need to make any specific improvements or something else. The authors provided the extended introduction, detailed description of materials and method, as well as the obtained results, their discussion and conclusions. Their study is supported well by many experiment results and deep analysis.     5. Are the conclusions consistent with the evidence and arguments presented and do they address the main question posed?   The conclusions are consistent well with the evidence and arguments presented in the article, and they address the main question posed.     6. Are the references appropriate?   All the article references are appropriate, and most of them are dated within the last 5 years.     7. Please include any additional comments on the tables and figures.   All the tables and figures are appropriate. They show well the research and experiment details and results. The first letter in ‘metric 1’, ‘metric 2’ and ‘metric 3’ should be capitalized in the captions of Figure 10, 11 and 12, respectively.     Also, I would like to advise the authors:   - to actualize the access date to the internet source specified in ‘Funding’;   - to capitalize the first letter in ‘metric 1’ in Line 525.   After detailed considering the paper, I have found that the results obtained are new and significant for the field. The paper is written well and ready for publication in the present form.   Finally, I conclude that the paper should be accepted for publication. Also, I would like to wish the authors further success in their research.

Author Response

Dear Reviewer,

Thank you very much for reviewing of our paper, and for your comments and detailed suggestions helped us to improve our paper. We hope that we have understood them well. In accordance with your comments and suggestions as well as suggestions of other reviewers, we have improved our manuscript as much as possible within the allotted time. All the changes/improvements are also highlighted in the Highlighted Changes file.

Reviewer 2:

  1. What is the main question addressed by the research? In the reviewing paper, the authors have considered the trusted operation of cyber-physical processes based on assessment of the system’s state and operating mode, and presented a method for detecting anomalies in the behavior of a cyber-physical system based on analysis of the data transmitted by its sensory subsystem. 2. Do you consider the topic original or relevant in the field? Does it address a specific gap in the field? I consider the topic original and relevant in the field. It addresses the solution of a very important problem of increasing the level of trust in the operating modes and processes of cyber-physical systems. Cyber-physical systems include a wide range of devices, ranging from sensors to artificial intelligence. The cyber-physical system combines various elements. Ensuring the security of such systems requires new solutions.     3. What does it add to the subject area compared with other published material?   The authors have proposed and investigated a method for detecting anomalies in the behavior of a cyber-physical system based on analysis of the data transmitted by its sensory subsystem. Their method allows quick detecting of failures with high accuracy. Compared to existing methods, the authors applied several metrics at once to assess the degree of trust in the process. They evaluated the process based on the analysis of changes in cyber-physical parameters. After determining the specific parameters, the authors determine whether the change in the parameter goes beyond the boundaries of the confidence interval. Moreover, the authors define several options for calculating the boundaries. Next, the authors determine whether an increase in the parameter is observed. Their method is quite flexible and takes into account the dynamics of the system.     4. What specific improvements should the authors consider regarding the methodology? What further controls should be considered?   There is no need to make any specific improvements or something else. The authors provided the extended introduction, detailed description of materials and method, as well as the obtained results, their discussion and conclusions. Their study is supported well by many experiment results and deep analysis.     5. Are the conclusions consistent with the evidence and arguments presented and do they address the main question posed?   The conclusions are consistent well with the evidence and arguments presented in the article, and they address the main question posed.     6. Are the references appropriate?   All the article references are appropriate, and most of them are dated within the last 5 years.     7. Please include any additional comments on the tables and figures.   All the tables and figures are appropriate. They show well the research and experiment details and results. The first letter in ‘metric 1’, ‘metric 2’ and ‘metric 3’ should be capitalized in the captions of Figure 10, 11 and 12, respectively.    

Response to Reviewer 2:

The first letters in ‘metric 1’, ‘metric 2’ and ‘metric 3’ have been capitalized in Figure 10, 11, and 12 captions.

Reviewer 2:

Also, I would like to advise the authors:   - to actualize the access date to the internet source specified in ‘Funding’;   - to capitalize the first letter in ‘metric 1’ in Line 525.   After detailed considering the paper, I have found that the results obtained are new and significant for the field. The paper is written well and ready for publication in the present form.   Finally, I conclude that the paper should be accepted for publication. Also, I would like to wish the authors further success in their research.

Response to Reviewer 2:

The access date to the internet source specified in ‘Funding’ has been actualized.

The first letters in ‘metric 1’ in Line 525 has been capitalized.

Reviewer 3 Report (New Reviewer)

1. It is critical to justify the research methodology, e.g. the author's preference of probability theory over fuzzy logic or other approaches. At least one alternative methodology should be mentioned in the Introduction or in the Discussions.

2. It is not clear from the paper whether selected threshold levels can be later customized by the operator or they will be hardcoded and embedded in the firmware.

3. There is a possible minor discrepancy between the abstract and the main body of the paper. In the abstract, three states are given: (1) trusted, (2) undetermined, and (3) unstable or under some form of stress. In paragraph 2.2, four states are given: stable, uncertain, unstable, and under-stress. Uniform terminology should be used across the paper.

4. Please, consider minor changes to the text of the abstract to incorporate the assumptions made in paragraph 2.2.

Abstract:

If the score is higher than 0.5, it means the system is in a trusted state. If the score is between 0.4 and 0.5, it means the system is in an indeterminate state. If the trust score tends towards zero, then this state can be interpreted as unstable or under some form of stress due to a system failure or deliberate attack.

vs.

Paragraph 2.2:

Since the experimental study shows that during normal flight some deviations from the course are possible, due to wind or GPS inaccuracies, we have set an acceptable deviation of 10–15% and thresholds of 0.4–0.6 for the cumulative function.

Author Response

Dear Reviewer,

Thank you very much for reviewing of our paper, and for your comments and detailed suggestions helped us to improve our paper. We hope that we have understood them well. In accordance with your comments and suggestions as well as suggestions of other reviewers, we have improved our manuscript as much as possible within the allotted time. All the changes/improvements are also highlighted in the Highlighted Changes file.

Reviewer 3:

  1. It is critical to justify the research methodology, e.g. the author's preference of probability theory over fuzzy logic or other approaches. At least one alternative methodology should be mentioned in the Introduction or in the Discussions.

Response to Reviewer 3:

We considered a comparison with a multi-criteria approach. But in accordance with your suggestion, we have made the following addition in before the last but one paragraph in Introduction:

“Sabo and Cohen [28] present a decision-making method for flying around obstacles using fuzzy logic rules. Their method allows converting binary parameters into fuzzy logic parameters. The method is quite effective for such a task. Nevertheless, in our study, it is important for us to detect not only the presence of some changes, but also to under-stand the nature of the process in order to verify the significance of the changes with respect to trust in the process. The metrics we have proposed allow us to do this.

Sun et al. propose a method for analyzing the failure of the GPS system in UAVs [29]. Their method, according to their authors, can be trained during operation. However, the method requires a database for training. In this case, the data must be marked up for storage and a considerable volume of data is required to implement the method. By contrast, in our solution prior data is needed only to confirm the effectiveness of the method and to verify it.”

and 2 new references:

“[28] Sabo, C.; Cohen, K. Fuzzy logic unmanned air vehicle motion planning. Adv. Fuzzy Syst. 2012, 2012, pp. 1–14. https://doi.org/10.1155/2012/989051.

[29] Sun, R.; Cheng, Q.; Wang, G.; Ochieng, W.Y. A novel online data-driven algorithm for detecting UAV navigation sensor faults. Sensors 2017, 17, 2243. https://doi.org/10.3390/s17102243.”

Also, we have reorganized the introduction and added a justification that our method is more suitable for the problems solved in the article compared to the existing ones.

Reviewer 3:

  1. It is not clear from the paper whether selected threshold levels can be later customized by the operator or they will be hardcoded and embedded in the firmware.

Response to Reviewer 3:

The following description of setting thresholds to the algorithm have added in the end of Section Materials and Methods:

  1. Determine the current flight mode.
    • If the UAV is in the current flight mode and switches to a new flight mode, then determine whether this transition is legitimate according to the transition graph, as shown in Figures 2 and 3.
    • If the mode is legitimate and the transition to the mode is legitimate, then , if not, then it becomes 0.
  2. Determine the reliability of the functions performed in the current state.
    • From the flight task file it is necessary to take information about: altitude, flight speed, longitude and latitude, their maximum, and minimum values and waypoints.
    • Calculate the boundaries of the confidence interval using formulas from lines 1.1.1 and 1.1.2 in Table 2.
    • From line 1.2.1 in Table 2, calculate the function of the probability of falling into the confidence interval for each of the parameters.
    • If condition is met, then set the parameter to 1 and go to Step 3.
    • If condition is not met, determine the parameter for which this condition was not met and assign the value 0 to the parameter. Return the value of the parameter and go to Step 3 .
  3. Control the increase/decrease of the cyber-physical parameter (Metric 2).
    • Calculate the average value of the cyber-physical parameter for intervals using the formula from line 2.2 in Table 2.
    • Calculate the value of the cumulative Poisson function for each cyber-physical parameter using the formula from line 2.1 in Table 2.

3.2.1 If the current processes are , and , then , or  and goes to zero, then set the parameter to 1 and go to Step 4.

If the condition above is not met within  intervals, then set the parameter to 0 and return the parameter value, for which the condition is not met ( set is optional, default is 3), and go to Step 4.

If the current modes are = , , , , and , then set the parameter to 1 and go to Step 4.

If the condition above is not met within  intervals, then set the parameter to 0 and return the parameter value, for which the condition is not met ( set is optional, default is 3), and go to Step 4.

3.2.2 If the current processes are , and , then the processes change the confidence function (formula from line 2.3 in Table 2) for latitude and longitude calculated relative to the starting point. That is, it is necessary to take the coordinates of the point from which the UAV starts taking off, obtain the lower and upper bounds of the confidence interval, and then calculate the value of the function.

If the value for the parameters of altitude and velocity is > 0.6 and tends to 1, then assign a value of 1 and go to Step 4.

If  < 0.6 during the intervals , then assign a value of 0 and return the value of the parameter for which the condition was not met (set is optional, default is 3), and go to Step 4.

3.2.3 If the current processes are , and , then the process change confidence function ( formula from line 2.3 in Table 2) for latitude and longitude calculated based on the point to which the UAV is aiming (also taken from the flight task).

If the value for the parameters of altitude and velocity is < 0,4 and tends to zero, then assign a value of 1 and go to Step 4.

If  > 0.4 during the intervals , then assign a value of 0 and return the value of the parameter for which the condition was not met (set is optionally, default is 3), and go to Step 4.

  1. Calculation of the final value (Metric 3) of trust by Formula 3.1 in Table 2. (For the final trust value, the values of each metric for each parameter are taken, depending on whether the values calculated above tended towards 0 or 1.)
    1. If = 0, it is necessary to provide an emergency response.
    2. If > threshold value, it means that the state is stable. (The threshold value depends on how many metrics we allow to be zero. We will assume that if three or more metrics are equal to zero, then an anomaly occurs, so the threshold will be taken as 0.6. This value is optional and can be changed by the user when testing the algorithm.)
  • If = 0.6, the system is in an undefined state.
  1. If < 0.6, the systems is in an unstable state.

We used a floating window equal to three time intervals. If the size of the floating window is increased, then the speed of anomaly detection will decrease, if the size is decreased, then the accuracy will be reduced. This value can be changed depending on how often the values are received from the flight controller. In our case, the values are obtained three times per second. Based on this, a floating window was also defined. As the frequency of data collection increases, the window may change. These calculations do not affect system performance by themselves due to the presented architecture of the software application.”

Reviewer 3:

  1. There is a possible minor discrepancy between the abstract and the main body of the paper. In the abstract, three states are given: (1) trusted, (2) undetermined, and (3) unstable or under some form of stress. In paragraph 2.2, four states are given: stable, uncertain, unstable, and under-stress. Uniform terminology should be used across the paper.

Response to Reviewer 3:

We have corrected it substituting ‘undetermined’ by ‘uncertain’ in Abstract.

Reviewer 3:

  1. Please, consider minor changes to the text of the abstract to incorporate the assumptions made in paragraph 2.2.

Abstract:

If the score is higher than 0.6, it means the system is in a trusted state. If the score is between 0.4 and 0.5, it means the system is in an indeterminate state. If the trust score tends towards zero, then this state can be interpreted as unstable or under some form of stress due to a system failure or deliberate attack.

vs.

Paragraph 2.2:

Since the experimental study shows that during normal flight some deviations from the course are possible, due to wind or GPS inaccuracies, we have set an acceptable deviation of 10–15% and thresholds of 0.4–0.6 for the cumulative function.

Response to Reviewer 3:

We have substituted those 3 sentences in Abstract by the following sentences:

“If the score is higher than 0.6, it means the system is in a trusted state. If the score is equal to 0.6, it means the system is in an uncertain state. If the trust score tends towards zero, then the system can be interpreted as unstable or under stress due to a system failure or deliberate attack.”.

This manuscript is a resubmission of an earlier submission. The following is a list of the peer review reports and author responses from that submission.

Round 1

Reviewer 1 Report

This paper is using the confidence interval of poisson distribution to detect anomaly. The method is too fundamental and experiments has no comparison to any state-of-art methods. I would argue to at least major revision or maybe direct rejection.

Author Response

Dear Reviewer,

Thank you very much for reviewing of our paper, and for your comments and detailed suggestions helped us to improve our paper. We hope that we have understood them well. In accordance with your comments and suggestions as well as suggestions of other reviewers, we have improved our manuscript as much as possible within the allotted time. All the changes/improvements are also highlighted in the Highlighted Changes file.

Reviewer 1:

This paper is using the confidence interval of poisson distribution to detect anomaly. The method is too fundamental and experiments has no comparison to any state-of-art methods. I would argue to at least major revision or maybe direct rejection.

Response to Reviewer 1:

This study is aimed not so much at detecting anomalies as at determining the level of trust in processes and cyber-physical systems. We use a cumulative function rather than a Poisson distribution to detect anomalies and calculate trust. In addition, we use three metrics at once for a more flexible assessment of trust. During the revision, we have added the merits of this work and comparison with similar methods. The experimental study and conclusions were also expanded. Some other improvements have been done in accordance with suggestions of other reviewers. Also, we have modified the article title in accordance with the changes performed.

Reviewer 2 Report

With the large-scale application of UAVs, they are increasingly affected by cyber attacks. How to perform the identification of attacks is an important and meaningful problem, and the paper proposes a method to identify the anomalies  based on analysis of the data transmitted by its sensory subsystem. And a series of the confidence interval is proposed, But there are some issues that are not described very clearly:

1. In line 161-165

What is the meaning of the slight change? How to determine the degree is slight enough to be acceptable?

2. In Line 261—273

a).What is the different between the unstable state and under-stress state? 

Where the unstable state “ can go beyond the scope of the current process, and with a low probability can reach the goal” and the “under-stress state” can going beyond the scope of the current process, with a high probability cannot reach the goal?

B). Is the Action set A in the 4 different sate are same?

C). “probability of achieving the goal T = {t 0 , ..., t i } set by the task ..;” How to set the probability by the task? 

3. In equation(6), the paper proposed a sample confidence interval for the UVA, while how does the numerical value 0.5 0.6 are given?

4. In the Experimental Results part, why the interval of the 0.6–0.9 was used as the acceptable confidence interval? Is it relevant with the data-set? Or is it consistent for all data-set?

5. With the method proposed by the authors it is possible to determine the outlier, how to determine whether this outlier is caused by an attack or a component failure?

Author Response

Dear Reviewer,

Thank you very much for reviewing of our paper, and for your comments and detailed suggestions helped us to improve our paper. We hope that we have understood them well. In accordance with your comments and suggestions as well as suggestions of other reviewers, we have improved our manuscript as much as possible within the allotted time. All the changes/improvements are also highlighted in the Highlighted Changes file.

Reviewer 2:

With the large-scale application of UAVs, they are increasingly affected by cyber attacks. How to perform the identification of attacks is an important and meaningful problem, and the paper proposes a method to identify the anomalies based on analysis of the data transmitted by its sensory subsystem. And a series of the confidence interval is proposed, But there are some issues that are not described very clearly:

  1. In line 161-165

What is the meaning of the slight change? How to determine the degree is slight enough to be acceptable?

Response to Reviewer 2:

We have added the following text at the end of the second paragraph on page 7:

“Deviations within 0.5–1.5 meters are implied. In this case, measurements of change have been identified experimentally based on many trials, as well as the result of theoretical analysis.”.

Reviewer 2:

  1. In Line 261—273

a). What is the different between the unstable state and under-stress state?

Where the unstable state “ can go beyond the scope of the current process, and with a low probability can reach the goal” and the “under-stress state” can going beyond the scope of the current process, with a high probability cannot reach the goal?

Response to Reviewer 2:

We have added a new paragraph after the second paragraph on page 11:

“An unstable state may occur as a result of deviations in 1 or 2 parameters. Then the confidence value will be slightly lower than 0.5. This situation is possible in the case of some deviations in flight, for example, due to the wind, when the drone cannot maintain the desired speed or altitude. The state under stress occurs due to significant external factors, usually because of an attack. Then 3 or more parameters are changed, but as the study showed, all parameters change.”

Reviewer 2:

  1. In Line 261—273

B). Is the Action set A in the 4 different sate are same?

Response to Reviewer 2:

The set of actions may be the same, or it may differ. Here, the assessment of cyber-physical parameters is more important. On the example of a drone, it can fly forward in a given direction, but not at the same speed and height. Or maybe, for example, perform a U-turn and fly in the other direction. That is, the actions may differ, or they may simply not correspond qualitatively. That is why two metrics are introduced when evaluating processes.

Reviewer 2:

  1. In Line 261—273

C). “probability of achieving the goal T = {t 0 , ..., t i } set by the task ..;” How to set the probability by the task?

Response to Reviewer 2:

The probability of achieving the task is related to the degree of trust in the process. The more likely the process is trusted, the more likely the goal will be fulfilled. Trust score metrics are shown in Table 2.

Reviewer 2:

  1. In equation (6), the paper proposed a sample confidence interval for the UVA, while how does the numerical value 0.5 0.6 are given?
  2. In the Experimental Results part, why the interval of the 0.6–0.9 was used as the acceptable confidence interval? Is it relevant with the data-set? Or is it consistent for all data-set?

Response to Reviewer 2:

Threshold values do not depend on the parameter. A new paragraph has been added at the end of section 2:

“Thresholds are taken based on the definitions of probability theory and distribution types. The threshold for the cumulative function is chosen because if the parameter does not deviate from the mean or expected value, then the distribution is 0.5. If a deviation occurs, then the value tends to be 0 or 1. Since the experimental study shows that during normal flight some deviations from the course are possible, due to wind or GPS inaccuracies, we have set an acceptable deviation of 10–15% and thresholds of 0.4–0.6 for the cumulative function. To get into the confidence interval metric 1 should tend to be 1. If the threshold of 0.5 is crossed, then we can talk about trust. Usually state 0 is considered untrusted, and 0.5 is undefined, as also described by Keshavarz et al. [18].”.

Also, the following sentences have been added at the end of the third paragraph in section 3 on page 14:

“The purpose of calculating the confidence interval is to build such an interval based on the sample data so that it can be asserted with a given probability that the value of the estimated parameter is in this interval. In other words, the confidence interval with a certain probability contains the unknown value of the estimated quantity. The wider the interval, the higher the inaccuracy. We set the threshold above 0.5 because it is assumed that the UAV operates in an untrusted and uncertain environment, when it can be affected by a large number of factors. At the same time, the UAV is able to cope with factors such as wind and GPS inaccuracy and can operate in such conditions. We lay the possibility of deviations of 10–15% as acceptable. The thresholds were obtained experimentally. When calculating the confidence interval, we obtained a probability function of how far the current value of the parameter falls within the boundaries of the confidence interval. This value should tend to be 1. Accordingly, a number greater than 0.8 tends to be 1. We have seen values from 0.6–0.9; they tend to be 1.”

Reviewer 2:

  1. With the method proposed by the authors it is possible to determine the outlier, how to determine whether this outlier is caused by an attack or a component failure?

Response to Reviewer 2:

Yes, it can be determined depending on which parameters are analyzed and depending on which of them have changed. We have completed the experimental study. In the first experiment, a spoofing attack was carried out on the UAV, and in addition to the parameters presented, the number of satellites and the noise level changed, which indicates spoofing. In the second experiment, the flight parameters were changed, but the noise and the number of satellites were unchanged.

Reviewer 3 Report

The manuscript proposes an approach to detect and address failures and attacks occurring in UAVs at different working modes. The topic is interesting and the work is valuable but need major improvement.

The following comments should be taken into account:

1.       Why do the variables follow Poisson distribution? Are we dealing with the number of incidents in a certain period? Even in that case, how do the noise effects be modeled in the system? Since noises follow a gaussian distribution.

2.       The introduced confidence interval in the draft is suitable for univariate variables. However, it is more common in the literature to design detection strategies for multivariate features,

3.       How were the thresholds adopted in terms of the confidence intervals?

4.       The authors need to investigate different validation methods and provide a thorough explanation accordingly. Moreover, please compare previous research works to the proposed approach, particularly studies conducted on the same dataset.

5.       The authors claimed that their method could detect unforeseeable abnormalities, such as hitting a bird’s flock. Nevertheless, throughout the manuscript, I cannot witness explicit information in this respect.

6.       Please recheck the English writing; some parts are ambiguous, for instance, lines 126-130.  

Author Response

Dear Reviewer,

Thank you very much for reviewing of our paper, and for your comments and detailed suggestions helped us to improve our paper. We hope that we have understood them well. In accordance with your comments and suggestions as well as suggestions of other reviewers, we have improved our manuscript as much as possible within the allotted time. All the changes/improvements are also highlighted in the Highlighted Changes file.

Reviewer 3:

The manuscript proposes an approach to detect and address failures and attacks occurring in UAVs at different working modes. The topic is interesting and the work is valuable but need major improvement.

The following comments should be taken into account:

  1. Why do the variables follow Poisson distribution? Are we dealing with the number of incidents in a certain period? Even in that case, how do the noise effects be modeled in the system? Since noises follow a gaussian distribution.

Response to Reviewer 3:

First, the Poisson distribution is not used in its pure form, but a cumulative distribution function is used. The cumulative function is a probability that estimates that a variable will take on a value X less than or equal to x. Thus, we evaluate whether the current parameter will deviate from the average value or not. If it deviates significantly, then the cumulative function will tend to be 0 or 1, depending on whether the parameter decreases or increases. Secondly, at certain values, the Poisson distribution becomes close to normal, so it is more universal for use.

Reviewer 3:

  1. The introduced confidence interval in the draft is suitable for univariate variables. However, it is more common in the literature to design detection strategies for multivariate features,

Response to Reviewer 3:

We consider each parameter separately. In the future, this may make it possible to determine the nature of the impact on the system, depending on the type of impact. Since we end up evaluating a normalized parameter, the method can be worked out for multivariate features.

Reviewer 3:

  1. How were the thresholds adopted in terms of the confidence intervals?

Response to Reviewer 3:

A new paragraph has been added at the end of section 2:

“Thresholds are taken based on the definitions of probability theory and distribution types. The threshold for the cumulative function is chosen because if the parameter does not deviate from the mean or expected value, then the distribution is 0.5. If a deviation occurs, then the value tends to be 0 or 1. Since the experimental study shows that during normal flight some deviations from the course are possible, due to wind or GPS inaccuracies, we have set an acceptable deviation of 10–15% and thresholds of 0.4–0.6 for the cumulative function. To get into the confidence interval metric 1 should tend to be 1. If the threshold of 0.5 is crossed, then we can talk about trust. Usually state 0 is considered untrusted, and 0.5 is undefined, as also described by Keshavarz et al. [18].”

Reviewer 3:

The authors need to investigate different validation methods and provide a thorough explanation accordingly. Moreover, please compare previous research works to the proposed approach, particularly studies conducted on the same dataset.

Response to Reviewer 3:

Unfortunately, other authors tend to use other metrics and assessment methods, and so it has not yet been possible to make a comparison. Nonetheless, we have also analyzed them and provided the following paragraphs after Figure 1:

“Trust in an automated system is an important factor in its development and com-missioning [17]. In addition, it is important to consider on what factors the credibility of the system is assessed. The level of performance exhibited by a system can affect the credibility of that system. Thus, the proper design of an autonomous system can be aided by measuring the trust in such systems. Lochner et al. added psychophysical factors to the assessment of trust in systems [17]. They took into account the state of the operator who controls the UAV as a trust factor by analyzing their galvanic skin response (GSR). In addition to the standard metrics, which are obtained by interviewing the operator, they collected skin conductivity data using a commercially available GSR (Shimmer Sensing) sensor. This device was attached to the wrist of the left hand, from which sensory electrodes were attached to the palmar surface of the index and middle fingers with two wires. The device transmitted data via Bluetooth to a laptop for data collection. This approach is quite interesting but not entirely applicable here because modern UAVs for the most part should be autonomous. In addition, modern UAVs have built-in functions in the flight controller, which already at the stage of its operation do not allow a person to fully influence the flight process, but only allow you to give certain sets of commands.

Keshavarz et al. [18] proposed a trust monitoring mechanism in which the ground station continuously monitors the behavior of the UAVs in terms of their trajectory, the energy they consume, as well as the number of tasks they have completed, and evaluates the UAV’s trust level to detect any abnormal behavior in real time. The simulation results show that the trust model can detect malicious UAVs that can be subject to various cybersecurity attacks, such as flood attacks, man-in-the-middle attacks, and real-time GPS spoofing attacks. When modeling a GPS spoofing attack, the authors present a drone flight graph. The graph looks like a rather chaotic trajectory, so most likely the simulation took place by substituting random coordinates. Such a simulation is not close to a real experiment and may give an inaccurate idea of the quality of the protection system. In addition, the group is monitored by the control center. Such a system in itself carries many shortcomings and can be easily compromised and destroyed by attacking the center.

Barka et al. proposed a new trust-monitor-based communication architecture for Flying Named Data Networking (FNDN) [19]. First, monitor nodes are selected based on their trust and stability. The monitors then become responsible for propagating the data packets to avoid the broadcast storm problem. At the same time, intermediate UAVs choose whether to validate data or not, following their subjective opinion about the behavior of their manufacturer and thus reducing computational complexity and latency. Simulation results show that this work can support security levels in excess of 80% dishonesty detection rate. The main idea is to establish trust between interacting UAVs and to select the most reliable UAV (called a monitor) with a high probability of storing data and having enough power for the rest of its mission. The direct trust score is calculated using the number of legal (L) and malicious (M) interactions between two UAVs i and j. An action is considered legitimate or malicious based on a measure of similarity between an effective action taken by UAV j and the same action modeled from the perspective of UAV i. Indirect trust is calculated based on recommendations from trusted UAVs about other unknown/known UAVs. Their study is based on the control of UAV actions, but they describe the categories of actions poorly. In addition, the presence of monitor nodes carries additional threats, since there are attacks that are aimed at breaching trust. In addition, their basic question is about the lifespan of the UAV network and how to assess trust if the UAV has just arrived in the group. On the one hand, it will have a lot of energy, but on the other hand, it will not have any historical recommendations.

A new context-sensitive trust-based solution for distinguishing between intentional and unintentional UAV anomalous behavior has been proposed by Barka et al. [20]. The method simultaneously establishes trust between UAVs and evaluates the current context in terms of UAV energy, mobility pattern, and queued packets. When evaluating confidence, the UAV evaluates the buffer occupancy, energy, and mobility patterns of the UAV. Thereafter, if the system detects that any adjacent UAVs have inadvertently dropped packets, it adds a confidence correction factor to the overall inter-UAV trust calculation. To distinguish unintentional from intentional dishonesty, the authors used three metrics in this work: (i) drops due to limited buffer space and data freshness, (ii) drops due to lack of power, and (iii) drops due to the mobility models of the selected forwarder. To evaluate the state of the buffer and decide if the drone is unintentionally dropping packets because its buffer is full, the average number of packets received and transmitted over a time interval is calculated. To assess the mobility of the UAV and decide whether the UAV is unintentionally dropping packets, the channel stability index is calculated. The advantage is in normalizinge the parameters and bringing them to values from 0 to 1. However, the main calculations are related to the transmitted and received packets and the communication channel. Such methods will not be suitable for cases where the UAV operates autonomously, and it will also be difficult to detect a data spoofing attack.

Maalolan [21] considered the assessment of trust in UAVs in a way similar to our method herein. He also considers the behavior of UAVs in different modes. There is an assessment of the flight speed, the rate of climb and descent, the quality of flying around obstacles, and more. Maalolan also considers the need to estimate the growth rate of the parameter in certain regimes, as we also propose in our study. A feature of his work is evaluating raw data. In our study, we use normalization and evaluate confidence in terms of probability. In addition, Maalolan uses monitors for each mode and set of metrics, but the final value of trust in the system is not developed.

Thus, the state of the art of our study is as follows:

– The developed method for calculating trust levels allows the CPS to assess the trust in the processes that occur during processing, or flying in the case of UAVs, in autonomous mode. Such a system can be scaled for a group of UAVs, they can exchange their values, and also build the ratings and reputation of a neighbor.

– The developed method uses normalized parameter estimation. The assessment uses two metrics. One of the metrics allows determining whether the process goes beyond the established confidence interval, and the second determines the growth rate of the parameter. The growth rate of the parameter is important for certain processes, such as starting or stopping, while when the system performs the same type of monotonous actions, the process must be static. The two metrics are combined to produce an overall level of confidence in the process, making it easier to decide whether to respond.

– The approach allows us to evaluate the trust in the process, taking into account the operating mode of the system, this is an important aspect, because in each mode the device can behave differently. In addition, we directly monitor the change of modes and evaluate how much this change is trusted. Thus, considering trust in conjunction with the processes in the system and expressing trust not in the system, but in the process, allows recognition of emerging malfunctions. This allows error checking functions not to exclude the system from work, but to correct its behavior. Such an adjustment is very important for dynamic, autonomous systems such as UAVs.”.

Also, we have extended section Experimental Results providing new Figures 9-12 and the following text:

“Now consider a situation where an attack on the UAV was not carried out, but the anomaly that arose nonetheless led to the UAV falling. It is necessary to assess whether our approach could detect such an anomaly if it were installed on a UAV. In this scenario, the UAV system crashed due to unforeseen reasons. This failure triggered “Failsafe mode” when the vehicle encounters a problem during flight, such as loss of manual control, critically low battery, or an internal error. Thus, we are dealing with an unforeseen situation that caused a system failure, the UAV deviated from the flight path, and as a result, the UAV crashed. Figure 9 shows two UAV flight options in mission mode. In the first case, the UAV successfully completed the mission, flying through all the established points (Figure 9a). In the second case, the UAV crashed due to unforeseen circumstances. Figure 9b shows that the UAV initially began to deviate from the course and could not even reach the first point. At the same time, no spoofing attacks were carried out, the UAV was recorded on 17 satellites, and the jamming and noise indicators were normal.

Let’s analyze the results of calculations of metric 1. The probability of falling into the confidence interval for the estimated parameters is shown in Figure 10.

Figure 10 shows that the longitude initially did not fall into the confidence interval, and starting from the 31st time interval, the latitude does not fall into the confidence interval either. Initially, the drone may have been launched at the wrong point and shifted, but then it straightens the flight path, and at the moment starting from the 31st time interval, it finally moves away from the given trajectory. These calculations correlate with the real situation, as can be seen from Figures 10 the UAV first simply shifts from the green line, and then begins to maneuver. The situation of a normal flight is highlighted in blue. Here you can see that the coordinates completely match the expected ones. As for the flight speed, it also did not report the expected one. The advantage of monitoring the changes is that we already get a finite value of the probability of falling into the confidence interval. In fact, we can evaluate each parameter using one formula and get uniform values that we can unambiguously attribute to trusted or untrusted. The flight altitude begins to change in the area where the UAV flies with strong deviations from the route, and this is fixed using the first metric.

Consider the results of calculating metric 2, which are presented in Figures 11. The mission mode is arranged in such a way that the UAV flies according to the flight task. At the same time, the speed and altitude of the flight specifically in this case should not have changed, so the rate of growth of the parameter should be unchanged. If the speed and altitude of the flight from point to point changed, then at certain moments we would observe an increase in this parameter. In this experiment, the growth rate of the parameter should be constant, so the values should be from 0.4–06. This parameter allows you to estimate how much the deviation from the expected value occurred, if there is a value of 0.5, then there is no deviation at all, since we allow deviations of 10–15%, then thresholds are selected based on this.

Figure 11 shows that the green graph, which indicates a normal flight, is almost always within the normal range, but some deviations are possible for the flight speed, because for a small UAV, maintaining one speed is quite a challenge due to varying wind conditions. The rest of the parameters on the graphs are normal. As for a situation under the influence of stress, we observe deviations in all parameters. Deviations are associated with a sharp increase or decrease in parameters, which is due to ongoing anomalous events with the UAV.

Consider the resulting trust value, which is the third metric shown in Figure 12.

It can be seen from Figure 12 that when an emergency occurs, the value of trust falls. The UAV did not immediately start to fall, but the confidence assessment allows early detection that problems have begun, giving time to take corrective measures. In general, it can be said that as soon as the UAV began to deviate from the course, the level of confidence dropped significantly to zero. It would be possible to take control from the operator and correct the flight or return the UAV back. Nevertheless, this was not done in this specific case, and the UAV continued to fly further and crashed as a result of an unforeseen error.”.

Reviewer 3:

  1. The authors claimed that their method could detect unforeseeable abnormalities, such as hitting a bird’s flock. Nevertheless, throughout the manuscript, I cannot witness explicit information in this respect.

Response to Reviewer 3:

A flock of birds was mentioned just as a possible example. To avoid misunderstanding in section 4, we removed this sentence from the article.

The experimental study has been improved, and an exception has been added. A new paragraph has been added before the last paragraph in Section Conclusion:

“As shown by the experimental study, this method makes it possible to determine not only the case when an attack is carried out on the UAV, but also other unforeseen situations. Recognition of such an anomaly is possible due to the fall of the confidence value to zero, as shown by both experiments. Although the value was not reduced to zero in the event of an attack, it reached 0.1. This is due to the fact that as a result of the attack, the spoofed UAV simply smoothly deviated from the route, and as a result of a different unforeseen circumstance, it fell. At the same time, the analysis of different groups of parameters can make it possible to determine what caused the failure. The advantage of our method is also that it allows analyzing trust in connection with the process or mode of operation of a cyber-physical system. This is important because a CPS can behave differently in different modes and different processes consume resources with different intensity, what is normal for one process will be an attack or anomaly for another process.”.

Reviewer 3:

  1. Please recheck the English writing; some parts are ambiguous, for instance, lines 126-130.

Response to Reviewer 3:

The paragraph has been rewritten:

“Some cyber-physical parameters can be obtained from different sensors, for exam-ple, the flight altitude value can be obtained from GPS or LIDAR. At the same time, such values obtained from different sensors may not coincide due to errors or inaccuracies in the operation of the systems [21]. Therefore, in this study, even though various devices may produce the same knowledge, in this study we will consider them as separate cyber-physical parameters. Thus, cyber-physical parameters enter the high-level control, where they are analyzed and determined in accordance with the expected parameters.”.

Other corrections in English usage have also been performed.

Round 2

Reviewer 1 Report

The technical contribution is too weak. I will like to reject it.

Reviewer 2 Report

The author did not answer the questions raised by the reviewers, and the setting of some key parameters in the article could not provide a basis, which made it difficult to evaluate the accuracy and effectiveness of the conclusions of the article

Reviewer 3 Report

The quality of the manuscript has been improved to a certain extent though some comments remain, as follows:

1.       Still a solid comparison with previous research studies has not been presented.

2.       Figure 9 is quite blurred. Please replace a higher resolution of that figure

3.       The authors did not address the reason for which multivariate approaches were not considered in this research.